# Detection Based Part-level Articulated Object Reconstruction from Single RGBD Image

**Yuki Kawana**[1]**, Tatsuya Harada**[1,2]
[1]The University of Tokyo, [2]RIKEN AIP
{kawana, harada}@mi.t.u-tokyo.ac.jp

## Abstract

We propose an end-to-end trainable, cross-category method for reconstructing multiple man-made articulated objects from a single RGBD image, focusing on part-level shape reconstruction and pose and kinematics estimation. We depart from previous works that rely on learning instance-level latent space, focusing on man-made articulated objects with predefined part counts. Instead, we propose a novel alternative approach that employs part-level representation, representing instances as combinations of detected parts. While our detect-then-group approach effectively handles instances with diverse part structures and various part counts, it faces issues of false positives, varying part sizes and scales, and an increasing model size due to end-to-end training. To address these challenges, we propose 1) test-time kinematics-aware part fusion to improve detection performance while suppressing false positives, 2) anisotropic scale normalization for part shape learning to accommodate various part sizes and scales, and 3) a balancing strategy for cross-refinement between feature space and output space to improve part detection while maintaining model size. Evaluation on both synthetic and real data demonstrates that our method successfully reconstructs variously structured multiple instances that previous works cannot handle, and outperforms prior works in shape reconstruction and kinematics estimation.

## 1 Introduction

Estimating object shape, pose, size, and kinematics from a single frame of partial observation is a fundamental challenge in computer vision. Understanding such properties of daily articulated objects has various applications in robotics and AR/VR.

Shape reconstruction of daily articulated objects is a challenging task. These objects exhibit a range of shapes resulting from different local part poses. More importantly, they display significant intra- and inter-category diversity in part configurations, including variations in part counts and structures. These factors together contribute to an exponentially increasing shape variation. Previous works have addressed this issue by either limiting a single model to target objects with a single articulated part [15] or employing multiple category-level models [28] to accommodate varying part counts. These approaches first detect each instance, and then model the target shape in an instance-level latent space, primarily employing A-SDF [38] for shape learning. A-SDF maps the target shape into an instance-wise latent space, and then a shape decoder outputs the entire shape of the instance. However, this approach is limited when dealing with varying part counts and structures, as the shape decoder must handle an exponentially increasing number of shape variations due to different part layout combinations [43, 9] in addition to local part poses. Consequently, addressing this variety with a single model remains a complex and unsolved task.

In this paper, we address this complexity through our novel detect-then-group approach. Our key observation is that daily articulated objects consist of similar part shapes. For example, regardless

37th Conference on Neural Information Processing Systems (NeurIPS 2023).

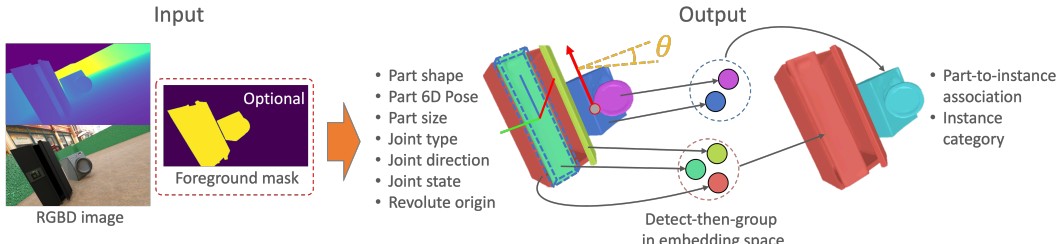

Figure 1: Our detection-based approach estimates part-level shape, pose, and kinematics as joint parameters. It also recovers parts-to-instance associations to handle multiple instances with various part structures and counts.

of the number of refrigerator doors, each door typically has a similar shape, and the base part may share similarities with those from other categories, such as storage units. By detecting each part and then grouping them into multiple instances, we provide a scalable and generalizable approach for handling diverse part configurations of daily articulated objects in a scene.

Based on this concept, we propose an end-to-end detection-based approach for part-level shape reconstruction. Building upon 3DETR [37] as an end-to-end trainable detector backbone, given a single RGBD image with an optional foreground mask, our model outputs part shape, pose, joint parameters, parts-to-instance association, and instance category. Our approach employs a novel detect-then-group approach. It first detects parts and applies simple clustering of parts into instances based on learned part embeddings' distance, in contrast to the previous works using additional instance detection module [28, 15]. An overview of our approach is shown in Fig 1. However, we found that detection-based shape reconstruction is prone to false positives for articulated objects' thin and small parts with little overlap, which is hard to remove by NMS. Also, articulated objects often have parts of varied sizes and scales, making training with a single-shape decoder challenging. Additionally, increasing model size by end-to-end training from detection to reconstruction makes simply enlarging the model size to improve performance undesirable. To address these challenges, we propose:(1) kinematics-aware part fusion to reduce false positives and improve detection accuracy; (2) anisotropic scale normalization for various part sizes and scales in shape learning; (3) and an output space refiner module coupled with a model-size balancing strategy with decoder for improved performance while keeping the model size. We evaluate our method on the photorealistically rendered SAPIEN [58] dataset, and our approach outperforms state-of-the-art baselines in shape reconstruction and joint parameter estimation. Furthermore, the model trained only on synthetic data generalizes to real-world data, outperforming the state-of-the-art methods on the BMVC [36] dataset.

Our contributions can be summarized as follows: (1) a novel part-level end-to-end shape reconstruction method for articulated objects from a single RGBD image; (2) a novel detect-then-group approach that simplifies the pipeline; (3) addressing detection-based reconstruction challenges with kinematics-aware part fusion, anisotropic scale normalization, and a refiner module coupled with model-size balancing; (4) superior performance on the SAPIEN [58] dataset, with the ability to generalize to real-world data from the BMVC [36] dataset.

## 2 Related Work

**Articulated shape reconstruction**   A number of works have focused on human subjects using single image input [46, 45] and utilize category-specific templates to recover deformation from a canonical shape [29, 64, 3, 63, 25]. Recent research has delved into recovering articulated shapes with unknown kinematic chains from sequences of a target in motion [41, 60]. These works make category-level assumptions about kinematic structures, with targets in observations sharing common kinematic structures. Their main focus is on natural objects such as humans and animals. In contrast, our interest is in reconstructing the shape of multi-category, daily man-made articulated objects with diverse kinematic structures using a single model. Recent years have seen the emergence of methods specifically targeting man-made articulated objects [56, 23, 53], and taking single-frame input [38, 28, 15, 24]. However, these models are constrained by either a predefined number of parts per category or the necessity of multiple models for each combination of categories and part counts. Consequently, they are unable to scale to a wide array of real-world articulated objects with varying part counts using a single model. Our approach addresses this limitation.

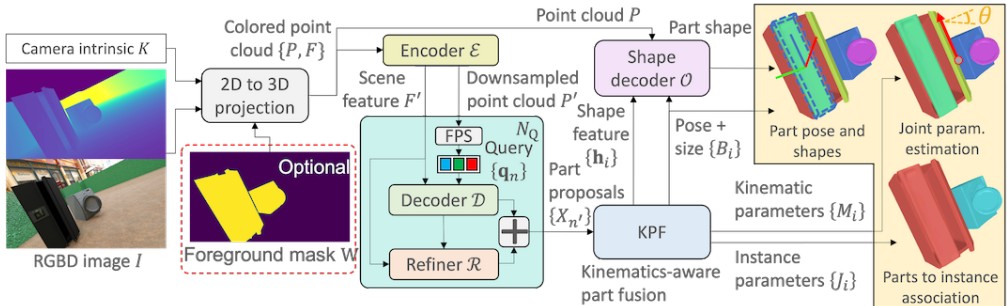

Figure 2: Overview of the pipeline. The input RGBD image is projected to a colored point cloud. The encoder $\mathcal{E}$ extracts scene features and a downsampled point cloud. The decoder $\mathcal{D}$ outputs a set of part proposals $\{X_n\}$ from part queryies $\{\mathbf{q}_n\}$. The refiner $\mathcal{R}$ estimates the residual of part pose and size $\Delta B$ and joint parameter $\Delta A$ for refined $\{B, A\}$. At test time, the inference is run $N_Q$ times independently to densely sample part proposals as $\{X_{n'}\}$. KPF removes false positives in $\{X_{n'}\}$ by using kinematics-aware IoU (kIoU) to refine the prediction further. The part shape is reconstructed by the implicit shape decoder $\mathcal{O}$.

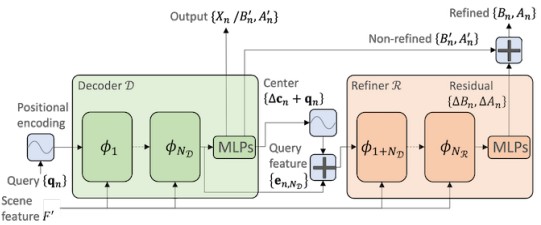

Figure 3: Architecture of refiner $\mathcal{R}$

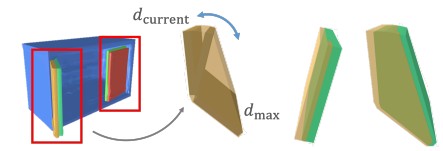

Figure 4: Illustration of (a) false positives, (b) Convex hull for kIoU, and (c) comparison of 3D box IoU and kIoU for overlapping parts.

**Pose and kinematic estimation of articulated objects** predominantly, existing research on pose and kinematic estimation of articulated objects has focused on estimation from sequences [16, 21, 57, 49], necessitating multiple frames of moving targets or interaction with the environment before estimation. Estimation from a single image has also been explored [27, 36, 28], but these methods are limited to predefined part structures. A few recent studies have proposed approaches without assumptions on part structure [22, 13]. However, these methods target single instances, requiring instance detection before part-level estimation. Moreover, their focus is limited to detection, pose and kinematic estimation, whereas our work aims for shape reconstruction in an end-to-end manner.

**Detection-based reconstruction** A large body of work exists combining detection and reconstruction for multiple rigid objects in diverse settings, such as indoor scenes [51, 39, 40, 61, 54, 14], tabletop environments [19, 20], and road scenes [2, 33]. Recent works target daily articulated objects [28, 16]. However, these methods predominantly rely on an instance-level detection approach. In contrast, our work pivots towards part-level detection to effectively handle a wide variety of part structures of real-world articulated objects.

## 3 Methods

### 3.1 Problem setting

Our method takes a point cloud $P$ with $N_P$ 3D points and color feature $F$ lifted from a single RGBD image $I$ of articulated objects and camera intrinsic $K$ as input. It outputs a set of parts $G = \{i \in [N]\}$ and non-overlapping subsets of all parts $\{G_l\}_{l=1}^{N_G}$ as $N_G$ instances, where $G = \bigcup_l \{G_l\}_{l=1}^{N_G}$. Note that we use the shorthand $\{*_i\}$ to denote an ordered set $\{*_i\}_{i=1}^{N}$ for brevity. For each part $i$, we estimate 6D pose and size $B \in \mathrm{SE}(3) \times \mathbb{R}^3$, part shape $\mathcal{O}$ as an implicit representation and kinematic parameters $M$. Our shape representation is an implicit function defined as $\mathcal{O} : \mathbb{R}^3 \to [0, 1]$ with isosurface threshold $\tau_{\mathcal{O}}$, where $\{\mathbf{x} \in \mathbb{R}^3 \mid \mathcal{O}(\mathbf{x}) > \tau_{\mathcal{O}}\}$ indicates inside the shape. The kinematic parameter $M$ consists of joint type $y \in \{0, 1, 2\}$ which represents fixed, revolute, and prismatic types, joint direction $\mathbf{a} \in \mathbb{S}^2$, 1D joint state $d_{\text{current}}$ and $d_{\text{max}}$ for the current pose and fully opened pose from

the canonical pose. We also predict the revolute origin $\mathbf{v} \in \mathbb{R}^3$ for the revolute part. We define the joint parameter as $A = \{\mathbf{a}, d_{\text{current}}, d_{\text{max}}[, \mathbf{v}]\}$, thus $M = \{y, A\}$. For each instance $l$, we estimate the instance parameter $J$ which consists of category $u$ and part association defined as $\delta_{li} = \mathbb{1}(i \in G_l)$, where $\mathbb{1}$ is an indicator function.

## 3.2 Detection backbone

Our detection backbone consists of a transformer-based encoder $\mathcal{E}$ and decoder $\mathcal{D}$ based on 3DETR [37]. The encoder comprises recursive self-attention layers encoding 3D points $P$ and color feature $F$ into downsampled 3D point cloud $P'$ of $N_{P'}$ points and $D_{F'}$-dimensional scene feature $F'$. Query locations $\{\mathbf{q}_n\}_{n=1}^{N_q} \in \mathbb{R}^3$ are randomly sampled using furthest point sampling (FPS) from $P'$. The decoder is composed of transformer decoder layers $\{\phi_k\}_{k=1}^{N_{\mathcal{D}}}$, considering cross-attention between queries and $F'$ and self-attention among queries. The decoder iteratively refines query features $\{\mathbf{e}_{n,k+1}\} = \phi_k(\{\mathbf{e}_{n,k}\}, F')$. Lastly, a set of part prediction MLPs decodes each refined query feature to produce output values. For clarity, the query index $n$ is omitted when possible.

## 3.3 Part representation

**Part pose and size** We predict part pose and size $B$ as a set of part center $\mathbf{c} \in \mathbb{R}^3$, rotation $\mathbf{R} \in \mathrm{SO}(3)$ and size $\mathbf{s} \in \mathbb{R}^3$ for each query. We predict $\mathbf{c}$ as an offset $\Delta\mathbf{c}$ from $\mathbf{q}$, added to the query coordinates, i.e., $\mathbf{c} = \mathbf{q} + \Delta\mathbf{c}$.

**Part shape** We employ a shared-weight, single implicit shape decoder $\mathcal{O}$ for performing part-wise shape reconstruction by taking point clouds around the detected regions where parts are identified. Given the diversity in shape and pose, and anisotropic scaling of the parts we focus on, it is challenging to learn shape bias with a single shape decoder. Therefore, we propose anisotropically normalizing the side lengths of the shape decoder's input and output to a unit scale to perform reconstruction. We define the input point cloud $P_{\mathcal{O}}$ as follows:

$$P_{\mathcal{O}} = \{(\mathbf{RS})^{-1}(\mathbf{p} - \mathbf{c}) \mid \mathbf{p} \in P, \max(|(\mathbf{RS})^{-1}(\mathbf{p} - \mathbf{c})|) \leq 0.5\}. \tag{1}$$

where $\mathbf{S}$ denotes diagonal matrix of scale $\mathbf{s}$. We define the output occupancy value at $\mathbf{x} \in \mathbb{R}^3$ as $o_{\mathbf{x}} = \mathcal{O}((\mathbf{RS})^{-1}(\mathbf{x} - \mathbf{c}) \mid P_{\mathcal{O}}, \mathbf{h})$, where $\mathbf{h} \in \mathbb{R}^{D_{\mathbf{h}}}$ is a part shape feature modeled by a part prediction MLP. Given that the input point cloud $P_{\mathcal{O}}$ includes background, the geometry of the target part shape can be ambiguous. To address this issue, we train the detector backbone and shape decoder end-to-end, by inputting $\mathbf{h}$ as shape geometry to the shape decoder so that $\mathbf{h}$ informs the shape decoder of the foreground target shape. We utilize a shape decoder architecture with a lightweight local geometry encoder [42] to spatially associate input points $P_{\mathcal{O}}$ with output occupancy values $o_{\mathbf{x}}$, which are both defined in normalized space.

**Part kinematics** We predict a 4-dimensional vector $y$ as a probability distribution over part joint types. This includes a 'background' or 'not a part' type for instances where predicted part proposals might not contain a part. The revolute origin $\mathbf{v} = \mathbf{q} + \Delta\mathbf{v}$ is predicted similarly to the part center $\mathbf{c}$, with an offset $\Delta\mathbf{v}$. Joint states $d_{\text{current}}$ and $d_{\text{max}}$ are modeled by separate part prediction MLPs for revolute and prismatic types, and we only supervise the output corresponding to the ground truth joint type. A single MLP is used for joint direction $\mathbf{a}$ for both revolute and prismatic types.

## 3.4 Instance representation

We form a collection of parts as a single instance $G_l = \{i \in G \mid \forall i, j \in G, i \neq j \Rightarrow \|\mathbf{z}_i - \mathbf{z}_j\| < \tau_{\mathbf{z}}\}$, where the distance between the $D_{\mathbf{z}}$-dimensional embeddings, $\mathbf{z}_i$ and $\mathbf{z}_j$, of each part pair within the group is kept below a specified threshold $\tau_{\mathbf{z}}$. We predict an $N_{\mathbf{u}}$-dimensional vector $\mathbf{u}$ as a probability distribution over instance categories per part. At test time, we predict the instance category by taking the category prediction with the highest confidence from the category predictions of the parts belonging to the same instance.

## 3.5 Refiner $\mathcal{R}$

Query features are iteratively refined in the decoder $\mathcal{D}$ in the feature space [37]. Increasing the number of decoder layers can enhance performance at the expense of a larger model size [37]. However, it is

crucial to avoid increasing the model size to enable efficient end-to-end training from detection to shape reconstruction. To improve model performance while maintaining the model size, we found cross-refining both output space and feature space to be effective. We introduce a refiner $\mathcal{R}$, which has an identical architecture with the decoder $\mathcal{D}$. The refiner $\mathcal{R}$ serves to refine the prediction in output space by reallocating a portion of decoder layers from decoder $\mathcal{D}$ to $\mathcal{R}$. Assuming there are $N_{\mathcal{D}+\mathcal{R}}$ decoder layers in the original model, we reallocate $N_{\mathcal{R}}$ decoder layers to the refiner $\mathcal{R}$. Consequently, $N_{\mathcal{D}} = N_{\mathcal{D}+\mathcal{R}} - N_{\mathcal{R}}$ layers are used for query feature refinement in decoder $\mathcal{D}$. The refiner $\mathcal{R}$ refines part pose and size $B'$ and joint parameter $A'$ from $\mathcal{D}$ by predicting residuals $\Delta B$ and $\Delta A$ to produce refined prediction $B$ and $A$, respectively. The architecture of the refiner $\mathcal{R}$ is shown in Fig. 3.

### 3.6 Kinematics-aware part fusion (KPF)

Articulated objects often consist of small and thin parts. Especially for unsegmented inputs, only a small portion of the point cloud represents such parts after subsampling. To ensure a sufficient number of queries cover such parts and their surrounding context, at test time, we randomize the sampling positions of the queries and independently carry out inference and NMS for the input $N_Q$ times, obtaining a collection of densely sampled part proposals from $N_Q$ inference runs. This process is termed Query Oversampling (OQ). However, OQ can increase false positives and degrade detection performance. To mitigate this, we propose Part Fusion (PF), inspired by Weighted Box Fusion (WBF) [48], to merge overlapping part proposals, thereby reducing false positives and improving detection accuracy. Unlike WBF, which fuses only 2D bounding parameters, PF fuses all the predicted parameters of part proposals, with an average weight defined as objectness $1 - \mathbf{y}_{\text{bg}}$. However, using IoU with 3D bounding boxes as overlapping metrics yields overly small values for thin structured parts, even when they are proximate. This results in PF failing to merge redundant parts, leading to false positives, as illustrated in Fig. 4 (a). To overcome this challenge, we propose a kinematics-aware IoU (kIoU) based on the observation that a redundant part pair exhibits significantly overlapping trajectories. To calculate kIoU, we construct a convex hull for each part in the pair using the 24 vertices of three bounding boxes based on size and canonical pose, current pose, and fully opened pose using the predicted part pose and size $B$ and joint parameter $A$, as show in Fig. 4 (b). Then, we calculate the IoU between the two hulls. The comparison between the 3D bounding box IoU and kIoU is depicted in Fig. 4 (c). Our overall method, termed Kinematic-aware Part Fusion (KPF), includes: (1) executing inference and NMS using kIoU to gather part proposals multiple times, (2) conducting PF using kIoU to update these proposals iteratively, and (3) removing proposals with low objectness. Note that KPF is non-differentiable and is disabled during training, with $N_Q = 1$. For further details, please refer to the appendix.

### 3.7 Set matching and training loss

**Set matching**  We base our end-to-end training of detection to reconstruction by utilizing 1-to-1 matching between part proposals and ground truth, using bipartite matching [5] for loss calculation. Similarly to [37], we define a matching cost between a predicted part and a ground truth part as $C_{match} = \lambda_1 \|\mathbf{B} - \mathbf{B}^{\text{GT}}\|_1 + \lambda_2 \|\mathbf{c} - \mathbf{c}^{\text{GT}}\|_1 - \lambda_3 \mathbf{y}_y + \lambda_4 (1 - \mathbf{y}_{\text{bg}})$, where $\mathbf{B}$ defines eight vertices of cuboid defined by part pose and size $B$, $\mathbf{y}_y$ defines the joint type probability given the ground truth label $y$, and $1 - \mathbf{y}_{\text{bg}}$ defines the foreground probability. Deviating from [37], we use the L1 distance of eight vertices of a cuboid instead of the GIoU [44] of cuboids in the first term to avoid the costly calculation of enclosing hull for 3D rotated cuboids.

**Training losses**  For each pair of prediction and ground truth, we define part loss as

$$\mathcal{L}_{\text{part}} = \frac{1}{N} \sum_i^N \|\mathbf{B}_i - \mathbf{B}_i^{\text{GT}}\|_1 + \|I - \mathbf{R}_i^T \mathbf{R}_i^{\text{GT}}\|_F^2 + \mathbb{E}_{\mathbf{x} \sim \mathbb{R}^3} \text{BCE}(o_{\mathbf{x},i}, o_{\mathbf{x},i}^{\text{GT}}) + \|d_{\max,i} - d_{\max,i}^{\text{GT}}\|_1 \tag{2}$$

$$+ \|d_{\text{current},i} - d_{\text{current},i}^{\text{GT}}\|_1 - \mathbf{a}_i^T \mathbf{a}_i^{\text{GT}} + \text{PL}(\mathbf{v}_i, \mathbf{v}_i^{\text{GT}}, \mathbf{a}_i^{\text{GT}}) + \text{CE}(\mathbf{y}_i, \mathbf{y}_i^{\text{GT}}) + \text{CE}(\mathbf{u}_i, \mathbf{u}_i^{\text{GT}})$$

All loss terms have equal weights. The first term is a disentangled L1 loss described in [47] to optimize $\mathbf{B}$. This loss is replicated three times by using only one of the predicted three components $(\mathbf{R},\mathbf{c},\mathbf{s})$ for $\mathbf{B}$, while replacing the other three with their ground truth values. We also found that directly optimizing rotation, as in the second term of the loss, leads to smaller rotation loss during training. BCE and CE denote binary cross-entropy and cross-entropy loss, respectively. PL denotes

| | F-Score@80% ↑ | F-Score@90% ↑ | CD@5% ↑ | CD@1% ↑ | IoU@25% ↑ | IoU@50% ↑ |
|---|---|---|---|---|---|---|
| A-SDF-GT [38] | 65.49 | 47.69 | 74.60 | 39.76 | 36.62 | 10.81 |
| A-SDF-GT-2 [38] | 68.81 | 52.55 | 75.91 | 43.58 | 38.59 | 10.43 |
| Ours-BG | 74.22 | **68.80** | 75.71 | **58.61** | 40.06 | 9.80 |
| Ours | **74.77** | 68.38 | **77.39** | 56.53 | **41.35** | **11.63** |

Table 1: Shape mAP results on SAPIEN [58] dataset.

the point-line distance between the revolute origin $\mathbf{v}_i$ and the ground truth joint axis defined by $\mathbf{v}_i^{\text{GT}}$ and $\mathbf{a}_i^{\text{GT}}$. For unrefined prediction $B'$ and $A'$, we define the same loss as $\mathcal{L}_{\text{part}}$ except for $o_{\mathbf{x}}$, $\mathbf{y}_i$ and $\mathbf{u}_i$ denoted as $\mathcal{L}'_{\text{part}}$. During training, we use ground truth $B$ for Eq.1 to avoid noisy prediction adversely affecting shape learning.

We also define instance loss $\mathcal{L}_{\text{instance}}$ for learning part-to-instance association with modified improved triplet loss [8] defined as:

$$\mathcal{L}_{\text{instance}} = \lambda_{\text{intra}} \sum_{i \in G} \frac{1}{|G_{l|i}|} \sum_{j \in G_{l|i} \setminus i} [\eta_{ij} - \tau'_{\mathbf{z}}]_+ + \frac{1}{N} \sum_{i \in G} [\max_{j \in G_{l|i} \setminus i} \eta_{ij} - \min_{j' \in G \setminus G_i} \eta_{ij'} + 3\tau'_{\mathbf{z}}]_+ \quad (3)$$

where $G_{l|i} = \{j \in G_l \mid \delta_{li} = 1\}$ and $\eta_{ij} = \|\mathbf{z}_i - \mathbf{z}_j\|$ denotes the L2 distance between the part-to-instance association embeddings of the $i$-th and $j$-th parts. The first term enforces the distance between the two embeddings of two different parts belonging to the same instance below $\tau'_{\mathbf{z}}$ and the second term ensures the distance between embeddings of parts belonging to different instances is larger than $3\tau'_{\mathbf{z}}$. However, computing all combinations of triplets for the second term is operationally complex for part-wise supervision. To streamline this, we instead opt to maximize the difference between the upper bound and the lower bound of inter- and intra-instance distances of the embeddings. In practice, we replace $\max$ and $\min$ with their soft approximations defined as $\text{LogSumExp}(\eta_{ij})$ and $-\text{LogSumExp}(-\eta_{ij'})$ for smooth gradient propagation. During inference, we use a threshold $\tau_{\mathbf{z}} = \frac{1}{2}(3\tau'_{\mathbf{z}} + \tau'_{\mathbf{z}})$ to determine if two parts belong to the same instance based on the distance between their embeddings.

The total loss we minimize is $\mathcal{L}_{\text{total}} = \mathcal{L}_{\text{part}} + \mathcal{L}'_{\text{part}} + \mathcal{L}_{\text{instance}}$. At training time, we use the same part prediction MLPs to predict part parameters at every layer in the decoder. We compute the $\mathcal{L}_{\text{total}}$ for each layer independently and sum all the losses. We only use the part parameter predicted from the last decoder layer at test time. Full loss formulation can be found in the appendix.

### 3.8 Implementation detail

The implementation of the transformer encoder-decoder architecture and the optimizer configuration are based on the publicly available code of 3DETR [37]. We employ a masked encoder architecture for $\mathcal{E}$ from [37]. The input point cloud to the encoder consists of $N_P = 32768$ points, which are subsampled to $N_{P'} = 2048$ points. The number of queries is set to $N_q = 128$ during training. At test time, we set it to $N_q = 512$, and with independent runs for query oversampling as $N_Q = 10$. The number of decoder layers for $\mathcal{D}$ and $\mathcal{R}$ are set to $N_{\mathcal{D}} = 6$ and $N_{\mathcal{R}} = 2$, respectively. For the evaluation of synthetic data, the foreground masks are inferred by a pretrained segmentation model using a ResNext50 [59] encoder and a DeepLabV3Plus [6] segmentation head with RGBD input. To evaluate real-world data, we automatically generate foreground masks using https://remove.bg, following [56]. The weights for the matching cost $\mathcal{C}_{\text{match}}$ were set to $\lambda_1 = 8, \lambda_2 = 10, \lambda_3 = 1, \lambda_4 = 5$. We set $\lambda_{\text{intra}} = 0.1$ for the intra-instance loss term of the instance loss $\mathcal{L}_{\text{instance}}$. We use the AdamW [31] optimizer with a base learning rate of 9e-4 and employ a cosine scheduler with nine epochs for warm-up. The training was performed using two A100 GPUs, each 40GB of GPU memory, and with a batch size of 26 for each GPU. The training took approximately two days. Please refer to the appendix for further details regarding the training process, hyperparameters, and network architecture.

## 4 Experiments

**Datasets** We evaluate our method on both synthetic and real-world data. We use the SAPIEN [58] dataset for synthetic data evaluation, following recent works on articulated shape reconstruction [24, 56, 15]. We select ten categories with representative articulation types, a sufficient number of CAD models, and various part structures across categories. We then randomly construct room

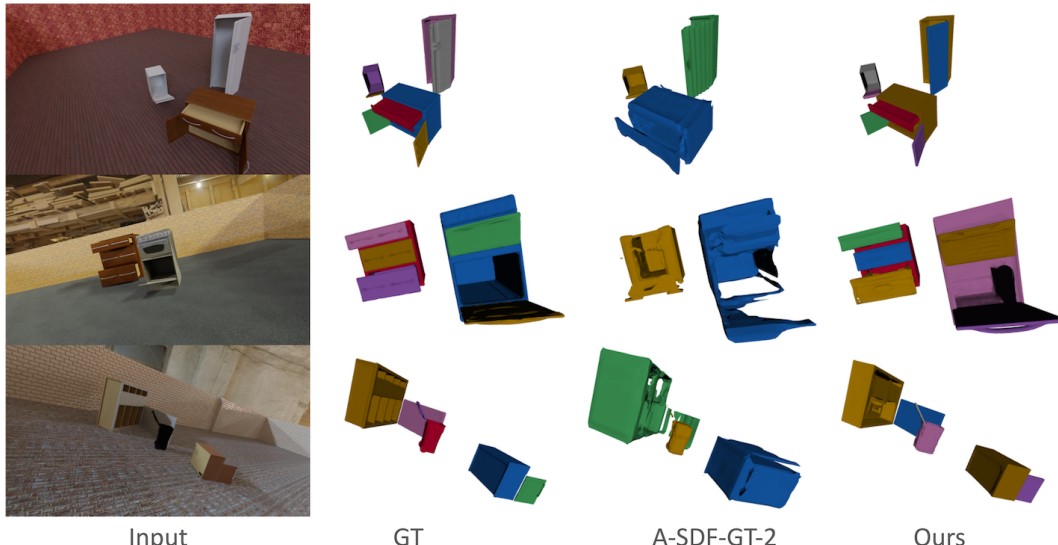

| Input | GT | A-SDF-GT-2 | Ours |
|---|---|---|---|

Figure 5: Qualitative results on SAPIEN [58] dataset.

| Category | Joint type | | Part count | | | Instance count | |
|---|---|---|---|---|---|---|---|
| | Rev. | Pris. | Most freq. | Min | Max | Train | Test |
| Dishwasher | ✓ | ✓ | 1 | 1 | 2 | 4997 | 1032 |
| Trashcan | ✓ | ✓ | 1 | 1 | 2 | 5037 | 1017 |
| Safe | ✓ | | 1 | 1 | 1 | 4795 | 1029 |
| Oven | ✓ | | 1 | 1 | 3 | 4951 | 1006 |
| Storage | ✓ | ✓ | 1 | 1 | 14 | 4907 | 973 |
| Table | ✓ | ✓ | 1 | 1 | 9 | 4790 | 999 |
| Microwave | ✓ | ✓ | 1 | 1 | 2 | 4832 | 934 |
| Frige | ✓ | | 2 | 1 | 3 | 5089 | 924 |
| Washing | ✓ | | 1 | 1 | 1 | 5042 | 995 |
| Box | ✓ | | 1 | 1 | 4 | 4823 | 979 |

Table 2: Overview of synthetic data from SAPIEN [58] dataset. Rev. and Pres. denote revolute and prismatic joints, respectively.

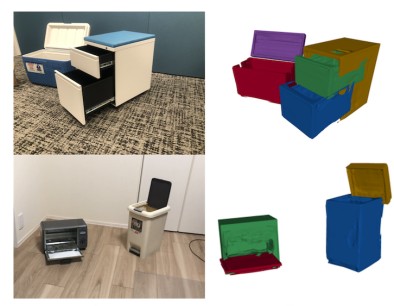

| Input | Ours |
|---|---|

Figure 6: Visualization on real-world data.

geometry (wall and floor) and place one to four instances per scene. Each instance is generated by applying random horizontal flips, random anisotropic resizing of side lengths, and random articulation of CAD models. The camera pose is sampled randomly, covering the upper hemisphere of the scene. Instances with severe truncation from the view frustum or occlusion with other instances are ignored during training and evaluation, ensuring that least one instance is visible in a view. For the training split, we randomize the textures of parts and room meshes. We use the original textures from the SAPIEN dataset for the test split. We generated 188,726 images for training and validation. Due to computational and time constraints, we used 20,000 images for training and kept the rest for validation usage. Also, we generated 4,000 images for the test split. Image size is $360 \times 640$ in height and width. The data overview is shown in Table 2. More details on data preparation can be found in the appendix.

For real-world data, we use the BMVC [36] dataset for quantitative evaluation. We use cabinet class which includes both prismatic and revolute parts and has the same part count and similar object shape as those used in our training and baseline models. The data contains two sequences of RGBD frames, capturing the same target objects with different part poses from various camera poses. We also test our method on RGB images taken with an iPhone X and depth maps generated from partial front views of the scene using Nerfacto [50].

**Metrics** For shape evaluation, following detection-based object reconstruction studies [40, 51, 16], we report mAP combined with standard shape evaluation metrics denoted as metric@threshold. We use F-Score [52], Chamfer distance (CD), and volumetric IoU (IoU) with multiple matching thresholds. More details on shape metrics can be found in the appendix. The shape mAP evaluation includes all the predicted parameters except for part kinematic parameters, serving as a comprehensive proxy for their quality. For part-wise detection with part pose and size, we use the average L2 distance

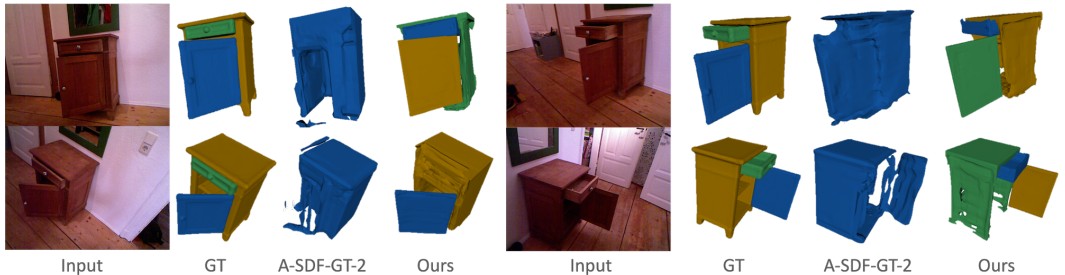

| Input | GT | A-SDF-GT-2 | Ours | Input | GT | A-SDF-GT-2 | Ours |

Figure 7: Qualitative results on the BMVC [36] dataset.

between the corresponding eight corners of bounding boxes between the ground truth and prediction as a matching metric of mAP, similar to the ADD metric [17], denoted as mAP@threshold. The distance is normalized by the ground truth part's diameter, with a proportion of the diameter used as the threshold. Since IoU-based metrics can yield values close to zero for thin structured parts, even when they are reasonably proximate and slightly off the ground truth, and do not consider part orientation, we use the above matching metrics to better analyze part-wise detection. For part kinematics evaluation, we evaluate the absolute error on joint state (State), Orientation Error (OE) for joint direction, and Minimum Distance (MD) for rotation origin following [22] for the detected parts with matched ground truth.

**Baselines** By default, we denote the proposed method that takes the foreground mask as 'Ours', and the model taking unsegmented input as 'Ours-BG'. To the best of our knowledge, no prior work operates on exactly the same problem setting as ours. For shape reconstruction, we benchmark against the state-of-the-art category-level object reconstruction method A-SDF [38], the closest to our approach. We follow the most recent work setup [15] in evaluation. A-SDF requires instance segmentation, pose, and scale to project the object from the camera to the object-centric frame. We implement it using all the required data with ground truth, denoted as A-SDF-GT following [15]. Given A-SDF's assumption of a fixed part count per model, as per a similar setup of [22], we train it on the dataset's most frequent part count per category. To account for multiple part counts in a category, for categories with more than one part count, we train the second model for the next most frequent count, resulting in a maximum of two models per category, termed A-SDF-GT-2. When the input instance has an untrained part count at test time, the model trained on the most frequent part count is used for A-SDF-GT-2. In the mAP evaluation, we give GT correspondence between the prediction and ground truth; thus, a false positive happens only when the shape metric is lower than the threshold for A-SDF. For kinematic evaluation, we compare our method with the state-of-the-art single-view part kinematic estimation method OPD [22]. Note that we input OPD with an RGBD image to align input modality and modify it to output joint state. More details on baselines can be found in the appendix.

## 4.1 Shape reconstruction

We show the shape mAP result in Table 1. Our method outperforms A-SDF [38] with ground truth in all metrics, and Ours-BG outperforms in the majority of metrics. We show the qualitative results in Fig. 5. Our models effectively reconstruct multiple instances with diverse part counts and structures, outperforming A-SDF which struggles with reconstructing articulated parts. We attribute our method's superior performance to its part-level representation, which facilitates easier learning of various part structures. Moreover, our shape-decoder is less affected by shape variations arising from the combination of part structures and poses. The qualitative results from novel viewpoints and additional visualizations can be found in the appendix.

## 4.2 Kinematic estimation

OPD [22] performs 2D detection evaluated by 2D IoU, while ours on 3D by L2 distance between 3D bounding box corners. To make kinematics estimation results of OPD and ours comparable, we experimentally choose the matching threshold of our mAP as 70% so that a similar number of detected parts with OPD's result with 2D segmentation mAP@50%. Then, we select the intersection of true positive detected parts by OPD and ours. The result is shown in Table 3. Our methods outperform OPD significantly. We attribute the reason to our method operating on 3D point clouds

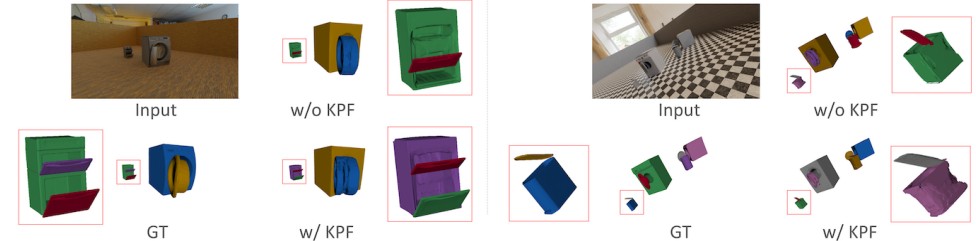

Figure 8: Qualitative results on kinematics-aware part fusion (KPF).

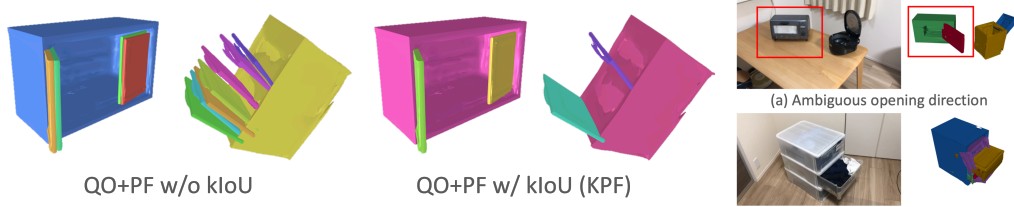

Figure 9: Qualitative results on kinematics-aware IoU (kIoU).

Figure 10: Failure cases.

while OPD is on 2D. Thus, our method is more robust to various textures and lighting conditions and makes it easier to reason about 3D kinematics. Note that our focus here is kinematics estimation after the *detection* step. Thus, superiority in detection performance is not our focus.

### 4.3 Ablation studies

In the following ablation studies, we validate each proposed component in a challenging setting where the input to the encoder has an unsegmented background using Ours-BG.

**Kinematics-aware part fusion** We show quantitative results in Table 4. Besides mAP, we also show precision considering false positives. QO denotes test-time query oversampling, and PF indicates part fusion. As a baseline, we first disable all components (w/o QO, PF, kIoU) using the same number of queries $N_q = 128$ during training and add each component one by one. Introducing our proposed kIoU on top of QO and PF outperforms the baseline in shape reconstruction mAP and part detection while preserving similar precision. We visualize the effectiveness of the KPF module in Fig. 8. In the provided comparison, 'w/o KPF' denotes disabling QO, PF and kIoU. We see that KPF enhances the detection and pose estimation of small parts. We also show the qualitative results on the proposed kIoU in Fig. 9. Applying QO and PF alone leads to false positives of thin parts, which kIoU effectively reduces. It indicates that the proposed KPF module improves overall detection performance while suppressing false positives.

**Anisotropic scaling and end-to-end training for shape learning** We validate the effect of anisotropic scale normalization and end-to-end training in shape mAP evaluation. In Table 5, w/o AS denotes using *isotropic* scaling by normalizing the *maximum* side length to one instead of *all* sides. w/o SF denotes not passing shape feature $\mathbf{h}$ but training shape decoder $\mathcal{O}$ separately. Disabling each component degrades performance. Especially disabling anisotropic scaling significantly drops the performance, as the single shape decoder is tasked to decode various sizes of target shapes.

**Ratio of decoder layers in the refiner $\mathcal{R}$** We investigate the relationship between the proportion of decoder layers in $\mathcal{R}$ and performance in shape mAP. The result is shown in Fig. 11. We vary the ratio of decoder layers in the refiner $N_{\mathcal{R}}/N_{\mathcal{D}+\mathcal{R}}$ from 0% to 75%. Allocating a portion (25%) of decoder layers to the refiner improves performance with the same number of decoder layers while reducing excessive decoder layers from the decoder degrades performance.

### 4.4 Real-world data

We verify the generalizability of our approach, which is trained only on synthetic data to real-world data. Here, we include foreground masks as inputs to mitigate the domain gap from background.

|          | State ↓          | OE ↓    | MD ↓     |
|----------|------------------|---------|----------|
| OPD [22] | 19.23°/17.10cm   | 29.68 ° | 38.11cm  |
| Ours-BG  | 4.30°/5.27cm     | 3.97°   | **6.43**cm |
| Ours     | **3.85°/4.06**cm | **3.66°** | 6.58cm |

Table 3: Joint parameter estimation results.

|                   | mAP@90 ↑ | Precision ↑ | F-Score@90% ↑ |
|-------------------|----------|-------------|---------------|
| w/o QO, PF, kIoU  | 38.87    | **52.29**   | 67.43         |
| w/o PF, kIoU      | 36.48    | 32.65       | 65.99         |
| w/o kIoU          | 40.78    | 49.64       | 66.24         |
| All (Ours-BG)     | **41.09**| 51.64       | **68.8**      |

Table 4: Ablation on KPF module.

|               | Fscore@80% | Fscore@90% | CD1@5%  | CD1@1%  | IoU@25% | IoU@50% |
|---------------|------------|------------|---------|---------|---------|---------|
| w/o AS        | 59.37      | 38.83      | 69.84   | 24.10   | 25.72   | 5.76    |
| w/o SF        | 71.04      | 60.49      | 73.83   | 43.14   | 25.17   | 0.64    |
| All (Ours-BG) | **74.22**  | **68.80**  | **75.71** | **58.61** | **40.06** | **9.80** |

Table 5: Ablation on shape learning.

We present the quantitative results on the BMVC [36] dataset in Table 6. As only one instance is present in the scene, we evaluate with shape metrics without mAP. The CD value is multiplied by 100. Our method outperforms A-SDF [38]. We observe reasonable generalization as shown in Fig. 7. Furthermore, we also show the qualitative result in Fig. 6 for scenes containing multiple instances. We observe reasonable generalization to real-world data.

# 5   Limitations

While our method makes significant strides in the daily articulated object reconstruction task, it does have several limitations. (1) Our method cannot handle cases where the instance boundary is not well defined, such as a door directly attached to a room. 3D reconstruction of room geometry with part-level shape reconstruction of articulated parts is an interesting future direction for scene reconstruction task. (2) The proposed approach currently does not consider the physical constraints between parts during training. Although the KPF module removes the redundant parts as post-processing, physically implausible false positives still occur as shown in Fig. 10 (a) for parts without overlapping trajectories. Introducing regularization, such as a physical violation loss [61], for such implausible configurations would alleviate this problem. (3) Our method does not explicitly consider the ambiguity in the opening direction of a part. As shown in Fig. 10 (b), the model fails to estimate the correct axis direction w.r.t. the base part for the closed part of the oven. This is because the dataset includes parts with different opening directions but similar shapes when closed. Such ambiguity can be addressed by explicit uncertainty modeling as in [1, 11], and integrating finer 2D visual cues for 3D reasoning [54, 26, 7] for localizing knobs and handles, which are informative for opening direction. Further discussions on limitations can be found in the appendix.

# 6   Conclusion

We presented an end-to-end trainable part-level shape reconstruction method for multiple articulated objects from a single RGBD image. We have demonstrated that our method successfully tackles the major limitation of previous works, which are unable to handle objects with various part counts using a single model, by employing a novel detect-then-group approach. We have also shown that the proposed kinematic part fusion (KPF) module effectively handles small parts as challenging targets while suppressing false positives for detection-based reconstruction. Our method outperformed the state-of-the-art baselines in shape reconstruction and kinematics estimation on the SAPIEN [58] dataset. Furthermore, on the BMVC [36] dataset and casually captured images, we demonstrated that the model trained on synthetic data reasonably generalizes to real-world data.

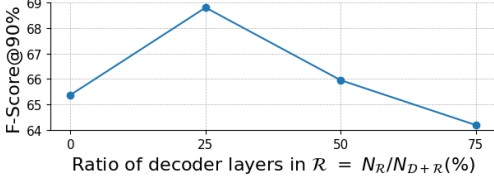

Figure 11: Ablation on refiner $\mathcal{R}$.

|                | F-Score ↑ | CD ↓  | IoU ↑   |
|----------------|-----------|-------|---------|
| A-SDF-GT-2 [38]| 83.25     | 1.73  | 12.26   |
| Ours           | **97.51** | **0.56** | **27.96** |

Table 6: Shape reconstruction results on the BMVC [36] dataset.

## Acknowledgments

We appreciate the members of the Machine Intelligence Laboratory for constructive discussion and their insightful feedback during the research meetings. This work was partially supported by JST Moonshot R&D Grant Number JPMJPS2011, CREST Grant Number JPMJCR2015 and Basic Research Grant (Super AI) of Institute for AI and Beyond of the University of Tokyo.

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

# A    Details on kinematics-aware part fusion (KPF)

In Algorithm 1, we present the details of kinematics-aware part fusion. We gather part proposals through $N_Q$ independent inference runs, achieved by randomly sampling query positions $\{\mathbf{q}_n\}$ with the furthest point sampling (FPS). The 'model' in the algorithm represents the decoder $\mathcal{D}$ and the refiner $\mathcal{R}$, and $\mathbf{y}_{\text{fg}}$ represents objectness, defined as $1 - \text{bg}$.

After part proposals are generated, we apply Non-Maximum Suppression (NMS) with 3D bounding box IoU and threshold by objectness $1 - \mathbf{y}_{\text{bg}}$ before applying NMS with kinematics-aware IoU (NMS-kIoU) for performance reasons. PF-kIoU denotes the part fusion (PF) process using kIoU, which is based on the Weighted Box Fusion (WBF) approach as described in [4]. The procedure is further detailed in Algorithm 2.

Unlike the conventional WBF, our algorithm fuses all parameters in the part proposal rather than only the 2D bounding box parameters. Furthermore, our algorithm iteratively runs until convergence, as demonstrated from line 11 to line 15 in Algorithm 1. $C$ represents clusters of overlapping part proposals identified by kIoU. We employ a simple arithmetic weighted average for all elements except the rotation matrix in a part proposal for fusion, denoted as weighted-average.

For the rotation matrix $\mathbf{R} \in \text{SO}(3)$, we initially derive a weighted average for the $3 \times 3$ matrices, labeled as $\mathbf{R}_{\text{wa}} \in \mathbb{R}^{3 \times 3}$. We then compute the weighted-averaged rotation matrix by minimizing the Frobenius norm to $\mathbf{R}_{\text{wa}}$. This is accomplished as $\mathbf{R} = \text{argmin}_{\mathbf{R}} \|\mathbf{R} - \mathbf{R}_{\text{wa}}\|_F$, following the 3D rotation library RoMa [4].

The value of $\mathbf{y}'_{\text{fg},n'}$ denotes the scaled objectness as a confidence score, considering the number of independent inferences $N_Q$. If the number of part proposals in a cluster is fewer than the independent inference runs $N_Q$, it indicates that only a limited number of independent inferences predict it. In such cases, we adjust the corresponding confidence score $\mathbf{y}'_{\text{fg},n'}$ by scaling it down by the ratio of the number of part proposals in the cluster ($|C[n']|$) to the number of independent inferences $T = N_Q$. Conversely, we scale up the confidence score if a cluster contains more part proposals than the independent inferences $N_Q$. For the initial run of part fusion (line 13 of Algorithm 1), we set $T = N_Q$ to fuse $N_Q$ inferences. We assign $T = 1$ for subsequent runs, assuming we apply part fusion to a single inference run fused by the previous part fusion.

**Hyperparameters**    The hyperparameter $\tau_{\text{IoU}}$ is set to 0.25, following the original code of 3DETR [37]. We set $\tau_{\text{obj}}$ to 0.25, representing the uniform probability of the three joint types plus background class, as outlined in Section 3. Additionally, we simplify the hyperparameter by setting $\tau_{\text{obj\_final}}$ equal to $\tau_{\text{obj}}$. The hyperparameter $\tau_{\text{kIoU}}$ is assigned a value of 0.5, double the $\tau_{\text{IoU}}$ value, to impose a stricter overlapping threshold and ensure similar trajectory part proposals are clustered together. We experimentally determine the values for $\tau_{\text{scaled}}$ and $\tau_{\text{count}}$, setting them to 0.1 and 3, respectively.

# B    Full Loss Formulation

During training, we use the same multi-layer perceptron (MLP) for part prediction at each decoder layer in the decoder $\mathcal{D}$. We calculate the total loss, $\mathcal{L}_{\text{total}}$, as defined in Section 3.7, independently for each layer and sum up these losses. The comprehensive loss formulation is as follows:

$$\mathcal{L}_{\text{total}} = \sum_{k=1}^{N_{\mathcal{D}}} \mathcal{L}_{\text{part},k} + \mathcal{L}'_{\text{part},k} + \mathcal{L}_{\text{instance},k}. \tag{4}$$

# C    Architecture

## C.1    Detection backbone

Our encoder and decoder architectures strictly follows the ones proposed in 3DETR [37]. We employ the masked-encoder from [37] as the encoder architecture for improved performance. We also employ the Fourier positional encoding, as proposed in [37], for the query embedding prior to input to the decoder. Please refer to [37] for comprehensive details and visualizations of the architectures.

---

**Algorithm 1** Kinematic-aware part fusion (KPF)

---

**Input:** Subsampled point cloud $P'$, scene feature $F'$
**Output:** Set of detected parts $\mathcal{X} = \{X_i\}$
 1: $\mathcal{X} \leftarrow \emptyset$
 2: **for** $N_Q$ times **do**
 3:     $\{\mathbf{q}_n\} \leftarrow \mathrm{FPS}(P')$
 4:     $\mathcal{X}' \leftarrow \mathrm{model}(\{\mathbf{q}_n\}, F')$
 5:     $\mathcal{X}' \leftarrow \mathrm{NMS}(\mathcal{X}', \tau_{\mathrm{IoU}})$
 6:     $\mathcal{X}' \leftarrow \{X_n \mid n \in [\mathcal{X}'], \mathbf{y}_{\mathrm{fg},n} > \tau_{\mathrm{obj}}\}$
 7:     $\mathcal{X}' \leftarrow \mathrm{NMS\text{-}kIoU}(\mathcal{X}', \tau_{\mathrm{IoU}})$
 8:     $\mathcal{X} \leftarrow \mathcal{X}' + \mathcal{X}$
 9: **end for**
10: count $\leftarrow 0$
11: **repeat**
12:     $\mathcal{X}_{\mathrm{old}} \leftarrow \mathcal{X}$
13:     $\mathcal{X} \leftarrow \mathrm{PF\text{-}kIoU}(\mathcal{X}_{\mathrm{old}})$
14:     count $\leftarrow$ count $+ 1$
15: **until** $\mathcal{X} = \mathcal{X}_{\mathrm{old}}$ or count $= \tau_{\mathrm{count}}$
16: $\mathcal{X} \leftarrow \{X_n \mid n \in [\mathcal{X}], \mathbf{y}_{\mathrm{fg},n} > \tau_{\mathrm{obj\_final}}\}$
17: **return** $\mathcal{X}$

---

---

**Algorithm 2** Part Fusion with kIoU (PF-kIoU)

---

**Input:** Set of part proposals $\mathcal{X}$
**Output:** List of updated part proposals $\mathcal{X}'$
 1: $C \leftarrow$ Empty list, $\mathcal{X}' \leftarrow$ Empty list
 2: Sort $\mathcal{X}$ in desceding order by objectness
 3: **for** $\forall X \in \mathcal{X}$ **do**
 4:     match $\leftarrow$ False
 5:     **for** $\forall X' \in \mathcal{X}'$ **do**
 6:         **if** $\mathrm{kIoU}(X, X') > \tau_{\mathrm{kIoU}}$ **then**
 7:             matched $\leftarrow$ True
 8:             $n' \leftarrow$ index of $X'$ in $\mathcal{X}'$
 9:             Append $X$ to $C[n']$
10:             $\mathcal{X}'[n'] \leftarrow$ weighted-average$(C[n'])$
11:             Break
12:         **end if**
13:     **end for**
14:     **if** not match **then**
15:         Append $X$ to $\mathcal{X}'$ and $C$
16:     **end if**
17: **end for**
18: **for** $\forall n' \in [\mathcal{X}']$ **do**
19:     $\mathbf{y}'_{\mathrm{fg},n'} \leftarrow \mathbf{y}_{\mathrm{fg},n'} \frac{|C[n']|}{T}$
20: **end for**
21: $\mathcal{X}' = \{X'_{n'} \mid n' \in [\mathcal{X}'], \mathbf{y}'_{\mathrm{fg},n'} > \tau_{\mathrm{scaled}}\}$
22: **return** $\mathcal{X}'$

---

## C.2 Part prediction MLPs

We adapt the box prediction MLPs outlined in [37] as our part prediction MLPs. Our part prediction MLPs retain the same architecture and are referred to as 'MLPs' in Fig. 3 of the decoder architecture. We employ 13 MLPs, each dedicated to a specific part proposal parameter. These parameters include the center offset from the query position $\Delta\mathbf{c}$, rotation $\mathbf{R}_{6D}$, size $\mathbf{s}$, current pose joint state for prismatic and revolute types $d_{\text{current}}^{\text{pris}}$ and $d_{\text{current}}^{\text{rev}}$, fully-opened pose joint state for prismatic and revolute types $d_{\text{max}}^{\text{pris}}$ and $d_{\text{max}}^{\text{rev}}$, joint axis $\mathbf{a}$, revolute origin offset from the query position $\Delta\mathbf{v}$, joint type $\mathbf{y}$, category $\mathbf{u}$, shape feature $\mathbf{h}$, and part-to-instance embedding $\mathbf{z}$. We utilize the 6D rotation representation [62] as the MLP's output representation $\mathbf{R}_{6D}$, subsequently converted to a $3 \times 3$ rotation matrix.

## C.3 Refiner

Our refiner architecture closely follows the decoder architecture, with MLPs outputting residual parameters. It follows the same architecture of the part prediction MLPs from the decoder, except for the dropout layers. We employ nine MLPs for residual predictions of center, rotation, size, prismatic and revolute types' current pose joint state, prismatic and revolute types' fully-opened pose joint state, joint axis, and revolute origin. For the rotation matrix and joint axis, the MLPs output 6D rotation representation which are then converted into $3 \times 3$ rotation matrices. These are updated by matrix product. The remaining parameters are updated through the addition of residuals. The same positional encoding is applied to the part center $\mathbf{c}$ to generate an embedding feature, which is then added to the corresponding refined query embedding $\mathbf{e}_{N_{\mathcal{D}}}$ from the decoder's final layer. This forms the new query feature input for the refiner, as depicted in Fig. 3.

## C.4 Shape decoder

Our shape decoder is based on the ConvONet architecture [42]. It consists of a local point encoder, a volume encoder, and an implicit shape decoder. The local point encoder quantizes input points to multiple 2D (tri-planes) or 3D grids and encodes points within each grid. The volume encoder then takes these locally encoded points and further encodes them into a 3D voxel feature. To decode the occupancy value at a 3D point $\mathbf{x}$, we sample the local feature $\mathbf{h}_{\text{local},\mathbf{x}}$ by interpolation on the 3D voxel feature at $\mathbf{x}$. The implicit shape decoder outputs the occupancy value $o_{\mathbf{x}}$ at $\mathbf{x}$, given $\mathbf{h}_{\text{local},\mathbf{x}}$ as input.

Our approach differs from the original ConvONet in that we concatenate the shape feature per part proposal $\mathbf{h}$ to the local feature to form $\mathbf{h}'_{\text{local},\mathbf{x}} = \mathbf{h}_{\text{local},\mathbf{x}} \oplus \mathbf{h}$. This serves as the learned part shape prior for the implicit shape decoder that reduces shape ambiguity due to the unsegmented per-part point cloud $P_{\mathcal{O}}$, as outlined in Eq. 1. We utilize the lightweight tri-plane architecture proposed in [42] for the volume encoder architecture. For more detailed information on the ConvONet architecture, please refer to [42].

## C.5 Foreground segmentation model

Our segmentation model incorporates a ResNext50 [59] architecture for the encoder with RGBD input and DeepLabV3Plus [6] for the segmentation head. We use the off-the-shelf implementation from [18].

# D Implementation details

## D.1 Architecture

Our approach strictly follows the hyperparameters of the masked-encoder described in [37]. We employ three self-attention transformer layers with self-attention masks between points in the point cloud, and each point only attends to others within a specified radius. The input point cloud to the encoder consists of $N_P = 32768$ points, which are subsequently subsampled to $N_{P'} = 2048$ points in the encoder. We set the scene feature dimension $D_{\mathbf{F}}$ a value of 256, following [37].

For the decoder, we maintain the hyperparameters from [37], except for the number of decoder layers $N_{\mathcal{D}}$. The decoder layers for the decoder $\mathcal{D}$ and the refiner $\mathcal{R}$ are set to $N_{\mathcal{D}} = 6$ and $N_{\mathcal{R}} = 2$, respectively. We match the query embedding dimension with the scene feature dimension for addition

and set the number of queries to $N_q = 128$ during training. For testing, $N_q$ is set to 512, unless stated otherwise, with independent runs for query oversampling (QO) for kinematics-aware part fusion (KPF) with $N_Q = 10$ independent inference runs.

As for the part prediction Multi-Layer Perceptrons (MLPs), we follow the box prediction MLP hyperparameters in [37], except for the number and output dimension of each MLP. The MLPs in the refiner for residual prediction strictly follow the part prediction MLPs hyperparameters in the decoder but without dropout. We set the dimension for shape feature $\mathbf{h} = 128$, and the dimension for part-to-instance association embedding $\mathbf{z} = 32$.

With respect to the ConvONet [42] architecture in the shape decoder, the local point encoder consists of five MLPs of width 128, and the volume encoder has three MLPs with widths of 16, 32, and 64. We employ the tri-plane version of the volume encoder for volume representation, using only the xy and yz planes to save GPU memory. In the implicit shape decoder, we employ four fully-connected layers with a hidden dimension of 128 and a leaky ReLU activation.

## D.2 Training details

We set the weights for the matching cost $\mathcal{C}_{\text{match}}$ at $\lambda_1 = 8, \lambda_2 = 10, \lambda_3 = 1, \lambda_4 = 5$. Each loss term in the total loss $\mathcal{L}_{\text{total}}$ carries equal weight. We use the AdamW [31] optimizer with a base learning rate of 9e-4 with a cosine scheduler down to a learning rate of 1e-6. The warm-up period is set to nine epochs, and mixed precision is used during training.

All models were trained on two A100 GPUs, each with 40GB GPU memory. We set the batch size to 26 per GPU, and it took approximately two days to complete 500 epochs. Weight decay was set at 0.1, with gradient clipping with an L2 norm of 0.1, as per [37].

We implemented on-the-fly sampling of 3D points and corresponding occupancy values to train the shape decoder. Half of these points are sampled uniformly from the space $[-0.5, 0.5]^3$. Additionally, a quarter of the points are sampled around the surface with a Gaussian-distributed random offset along the surface normal direction, with a standard deviation of 0.1. The remaining quarter is sampled with a standard deviation of 0.01. Each part has 128 points sampled for occupancy values.

In training the foreground segmentation model, we used the AdamW optimizer and a cosine scheduler, identical to the approach used in training the previous models. This model was also trained on two A100 GPUs with 40GB GPU memory and a batch size of 26 per GPU. Training took approximately one day for 500 epochs.

We add noise to the depth map during training, following the approach in [32]. We incorporate a depth map filter for the model tested on real-world data to mitigate the flying pixel effect on the object edge as suggested by [55]. This filter is also applied during testing for real-world data. Following the projection of the depth map to 3D points, we introduced random scaling within a range of $\pm 15\%$, and random rotation along the surface normal direction within $\pm 30°$. The point cloud was zero-centered for both training and testing. Color augmentation during training follows the original code of [37]. For the model using a foreground mask, we augment the mask by randomly displacing the foreground pixels around the border and performing successive dilation and erosion to simulate noise on inferred masks during training.

## D.3 Mesh generation

We sample occupancy values on $64^3$ voxel grids per part, then extract the surface mesh using the marching cubes method [30] at an isosurface level $\tau_{\mathcal{O}} = 0.3775 = \text{Sigmoid}(-0.5)$. For surface mesh reconstruction evaluation, we store an instance mesh as a union of part meshes and apply quadratic decimation [12] to reduce the number of faces to 10000. For volumetric IoU evaluation, where we evaluate the union of IoU for each part, we stored a part mesh and applied quadratic decimation to reduce the number of faces to 5000.

# E   Dataset Details

## E.1   SAPIEN [58] dataset

We choose ten categories offering prismatic and revolute joints as representative articulation types, adequate CAD models, and various part structures. The CAD models are randomly divided into training + validation splits (referred to as the trainval split) and test split, maintaining an approximate 8:2 ratio. No overlapping CAD model IDs exist between these splits.

All CAD models are manually aligned such that they stand upward along the y-axis and face the z-axis. We use BlenderProc [10] to generate synthetic data. We randomly create room geometry (wall and floor) in each scene and place one to four instances.

For the trainval split, we randomize the texture of the part mesh and room geometries. The textures for the part mesh are randomly applied from https://www.blenderkit.com, using the downloading script of BlenderProc [10]. We employ 2923 textures for this split. However, for the test split, we use the original texture from the SAPIEN [58] dataset.

The room geometry textures are downloaded from https://ambientcg.com using the BlenderProc script [10] and randomly applied. We use 353 textures for the trainval split and 85 textures for the test split, ensuring no overlap.

For environmental lighting, HDR maps downloaded from https://polyhaven.com are used. We employ 113 HDR maps for the trainval split and 29 for the test split, with no overlap.

Instances are created by applying random horizontal flips, random anisotropic resizing of side lengths, and random articulation of CAD models. The maximum joint state of the revolute joint defaults to $135°$ unless it touches the ground. We set the maximum joint state value to avoid ground contact if it does. We use the CAD model's maximum prismatic joint state value for prismatic joints, adjusted for rescaling.

This paper considers instances comprising one base part and at least one articulated part attached to it. Therefore, we fix articulated parts attached to another articulated part and ignore their joints, such as small prismatic buttons and rotating knobs on doors.

The scene is populated with instances by first sampling one to four categories uniformly, with replacement. Then, a CAD model is randomly sampled for each selected category, and its part pose, size, and yaw rotation are randomly augmented. The size distribution is shown in Fig. 13. The yaw rotation is limited to $\pm 90°$, ensuring instances face roughly forward. Instances are placed randomly in the scene, maintaining a minimum and maximum distance of 0.1m and 0.8m from other instances, respectively. This setup prevents instances from being scattered in the room.

We allow up to 10 sampling of camera poses in the same scene, limiting the camera position to the union of convex hulls of instances projected on the x-z plane (same as the floor) and a circular sector whose center is at the scene center and the arc is an angle between $\pm 80°$. This restriction, coupled with the yaw range limitation, discourages the self-occlusion of parts by an instance facing backward to cameras. We illustrate this idea in Fig. 13.

We use the BlenderProc API to sample camera poses against instances with a minimum distance to a camera from an instance of 0.5m. In total, we render 4,000 frames for the test split and 188,726 for the trainval split, randomly sampling 20,000 frames for training and allocating the rest for validation usage.

We exclude instances whose 2D silhouette is occluded more than 50% or truncated by 25%. Each frame is ensured to have at least one visible instance. The image size is $360 \times 640$ (height and width), and the field of view is set to $86.64°$.

For the generation of watertight mesh for the implicit shape ground truth in shape learning, we follow the instructions from [35] to generate part-wise watertight meshes. We then apply quadratic decimation [12] to the generated part mesh, reducing the face count to approximately 10,000 when combined as an instance.

| | Dishwasher | Trashcan | Safe | Oven | Storage | Table | Microwave | Frige | Washing | Box |
|---|---|---|---|---|---|---|---|---|---|---|
| Trainval | 33 | 47 | 24 | 14 | 266 | 60 | 10 | 33 | 13 | 20 |
| Test | 9 | 11 | 6 | 4 | 68 | 16 | 3 | 9 | 4 | 6 |

Table 7: Number of CAD models from SAPIEN [58] dataset used in trainval and test splits for our synthetic dataset.

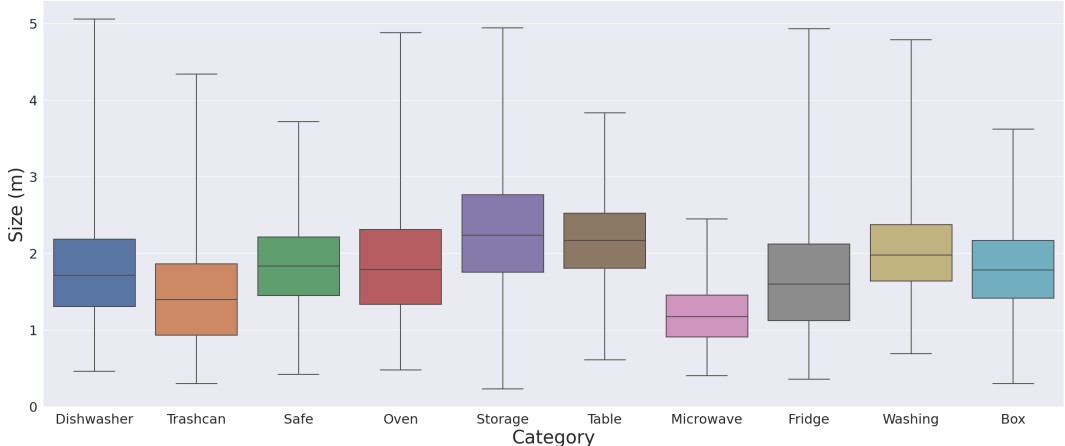

Figure 12: Size distribution per category after size augmentation applied to CAD models.

## E.2    BMVC [36] dataset

We use the cabinet class data from this dataset, which includes one prismatic part and one revolute part. The original data comprises two sequences of RGBD frames, with different part poses for each sequence and varied camera poses. The first sequence contains 1108 frames, while the second has 1119 frames. The image size is $480 \times 640$ (height and width).

Before feeding the data to the model, we apply a depth map filter to suppress the flying pixel effect on the object edge, as detailed in [55]. We extract the foreground point cloud using the silhouette of the projected CAD model on the image.

## E.3    Images taken by iPhone X

We record the partial front view of the scene as a short clip, extracting approximately 300 frames to train the neural rendering model Nerfacto [50] for depth map generation. Default training hyper-parameters are used for Nerfacto training. The image size is $540 \times 960$ (height and width). For our evaluation, foreground masks are automatically extracted using https://www.remove.bg unless specified otherwise. Note that we only use the point cloud lifted from a single RGBD image as input, not recovered from all frames.

# F    Details on baselines

## F.1    A-SDF [38]

**Data preparation**    We follow the description in the original A-SDF paper [38] for data preparation. After normalizing the global pose and scale, we ensure the base part's position does not change regardless of random articulation during training. In addition, for categories with multiple parts, we maintain consistent part ordering. We automate this by assigning part positions in canonical poses to a cell in the $5 \times 5 \times 5$ voxel, which discretizes the positions. The 1D flattened cell positions of the voxel determine the part order. To limit the training time, we randomly sample a maximum of 1500 instances per category for the training split.

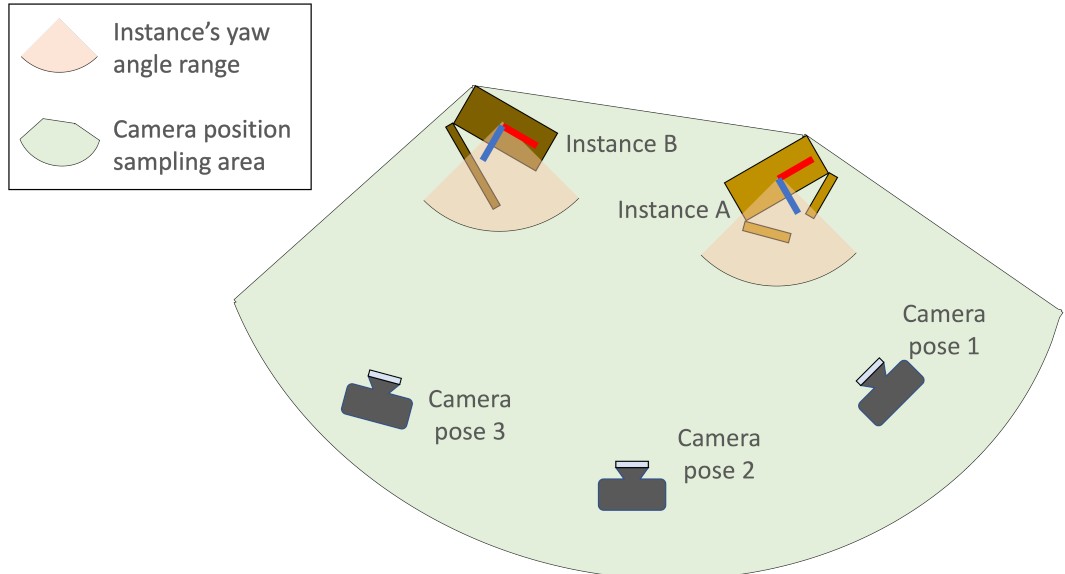

Figure 13: Illustration of yaw rotation range of an instance and camera position sampling area.

**Model**   The model implementation uses publicly shared author code. The original A-SDF only targets revolute parts. We normalize the prismatic joint state by the maximum joint state to fall within the $[0, 1]$ range. We then multiply this range by 135, ensuring joint states for revolute and prismatic parts are within the same range to extend the model for prismatic parts.

A-SDF is a category-level approach that works with a fixed number of parts, which becomes a problem when evaluated on dataset containing instances with varied part counts in a category. Similarly to our approach, OPD [22] can handle multiple part counts without assuming the predefined part number. They evaluate their method against baselines targeting a fixed number of parts by training the baselines for a category's most common part count.

We follow this protocol but aim to favor the baseline A-SDF more by training up to two models per category. The first model is for the most common part count, and the second, if applicable, is for the next most common part count. The second model is used when a category has more than two different part counts in the test split.

This setting differs from OPD's method, which only uses the most common part count. During testing, we use the model with the most frequent part count for instances with untrained part counts. We call the baseline model trained for the most frequent part count A-SDF-GT and the one using the next most frequent part count A-SDF-GT-2. The baselines and per category part counts, excluding the base part, are summarized in Table 8.

**Training**   We employ the publicly shared author code for training. During training, each part's articulation varies randomly from 0°to 135°, sampled at 5°intervals as suggested in [38]. We dynamically sample 3D points and their corresponding signed distance values from the randomly articulated watertight meshes of parts during training. The number of sample points and the ratio of uniformly sampled points to points near the mesh surface follow the original A-SDF paper [38]. A-SDF model training uses part label supervision and takes approximately ten days on a single V100 or A100 GPU per model.

**Mesh generation**   For mesh generation, we use the publicly shared author code as well. A-SDF assumes no background and normalized global pose and size for input. During testing, we use the ground truth instance segmentation mask for each instance in a scene to isolate instance-wise foreground points. We also normalize the depth map to the input space using the ground truth pose and size. The surface mesh is extracted using the provided implementation, which samples signed distance values on $64^3$ voxel grids and extracts surface mesh using marching cubes [30]. Additionally,

| | Dishwasher | Trashcan | Safe | Oven | Storage | Table | Microwave | Frige | Washing | Box |
|---|---|---|---|---|---|---|---|---|---|---|
| Most freq. | 1 | 1 | 1 | 2 | 1 | 1 | 1 | 2 | 1 | 1 |
| Next Most freq. | n/a | 2 | n/a | n/a | 2 | n/a | 2 | 1 | n/a | 4 |

Table 8: The most frequent and the next most frequent part count for each category that baseline A-SDF models are trained on.

we apply the quadratic decimation [12] to limit the number of faces to 10000. We then project the generated mesh back to the original scale and pose in the scene using the ground truth values.

### F.2 OPD [22]

**Data** Following the original paper [22] and the publicly shared author code and dataset, we generate ground truth data. The original data includes a semantic label of part type, one of drawer, lid, or door. However, our synthetic dataset does not have corresponding labels. Thus we replace the semantic label with the joint type. Consistent with the original paper, we exclude the fixed type part from the ground truth, targeting only the revolute and prismatic parts for detection.

**Model** We employ the OPDRCNN-C model from [22], which predicts kinematic parameters in camera coordinates. For input modality alignment, the model uses RGBD input. We have adapted the model to produce a continuous 1D joint state; instead of supervising the joint state prediction head with binary joint state supervision, we use continuous 1D joint state supervision. We duplicate the MLP head to separately predict joint states for revolute and prismatic joints. We supervise only the output corresponding to the ground truth joint type. Furthermore, we guide the joint axis prediction with reference to the floor's normal direction, as we experimentally found this improves detection performance.

**Training** We follows the author's implementation for training code, maintaining the same settings and hyperparameters for the OPDRCNN-C model.

**Testing** As described in Sec. 4, we match the ground truth using part segmentation evaluation with an IoU threshold of 50% following the author's implementation. We then evaluate the predicted joint parameters against the matched ground truth.

## G Category-wise shape mAP

We show category-wise shape mAP results in Table 9, 10, 11, 13, 14, 15, 16, 17, 18.

## H Additional results on the refiner $\mathcal{R}$

We evaluate the effectiveness of the refiner $\mathcal{R}$ in enhancing performance while maintaining a comparable model size. Table 19 shows the quantitative results on the shape mAP and the number of learnable parameters of the Ours-BG model, with the number of decoder layers in the decoder $N_{\mathcal{D}} = 6$ and in the refiner $N_{\mathcal{R}} = 2$. We compare this model against the model without a refiner and having the same total number of decoder layers $N_{\mathcal{D}} = 8$, and another model with a doubled number of decoder layers $N_{\mathcal{D}} = 16$ without the refiner. The result shows that using the refiner achieves comparable performance in shape mAP with the model with a doubled number of decoder layers $N_{\mathcal{D}} = 16$ with a smaller model size.

| | Fscore@80% | Fscore@90% | CD1@5% | CD1@1% | IoU@25% | IoU@50% |
|---|---|---|---|---|---|---|
| A-SDF-GT [38] | 59.71 | 43.14 | 62.79 | 29.64 | 29.64 | 0.00 |
| A-SDF-GT-2 [38] | 58.40 | 41.49 | 61.36 | 26.23 | 26.89 | 0.22 |
| Ours-BG | 85.73 | 83.93 | 85.94 | **80.72** | 41.75 | 0.15 |
| Ours | **88.28** | **84.67** | **91.14** | 76.00 | **42.01** | **0.33** |

Table 9: Shape mAP result of dishwasher category.

|  | Fscore@80% | Fscore@90% | CD1@5% | CD1@1% | IoU@25% | IoU@50% |
|---|---|---|---|---|---|---|
| A-SDF-GT [38] | 78.76 | 61.84 | 82.73 | 55.08 | 12.95 | 0.00 |
| A-SDF-GT-2 [38] | **81.45** | 70.83 | **84.13** | 62.78 | **21.24** | 0.00 |
| Ours-BG | 76.41 | 72.21 | 78.67 | 71.15 | 16.78 | **0.01** |
| Ours | 78.23 | **74.63** | 81.03 | **72.32** | 19.08 | 0.00 |

Table 10: Shape mAP result of trashcan category.

|  | Fscore@80% | Fscore@90% | CD1@5% | CD1@1% | IoU@25% | IoU@50% |
|---|---|---|---|---|---|---|
| A-SDF-GT [38] | 69.19 | 48.34 | **73.42** | 40.21 | 37.00 | 5.15 |
| A-SDF-GT-2 [38] | **69.53** | 49.83 | 73.31 | 41.24 | 38.14 | 5.61 |
| Ours-BG | 59.25 | **55.52** | 59.45 | **44.15** | **46.43** | 3.27 |
| Ours | 56.69 | 53.21 | 57.47 | 43.40 | 46.03 | **6.28** |

Table 11: Shape mAP result of safe category.

|  | Fscore@80% | Fscore@90% | CD1@5% | CD1@1% | IoU@25% | IoU@50% |
|---|---|---|---|---|---|---|
| A-SDF-GT [38] | 83.86 | 82.17 | 84.09 | 72.80 | **83.75** | **41.31** |
| A-SDF-GT-2 [38] | 83.86 | 82.05 | 84.09 | 71.90 | 83.52 | 40.86 |
| Ours-BG | 89.01 | 87.28 | 90.64 | **79.68** | 83.28 | 15.60 |
| Ours | **90.62** | **87.31** | **92.51** | 78.53 | 82.25 | 29.65 |

Table 12: Shape mAP result of oven category.

|  | Fscore@80% | Fscore@90% | CD1@5% | CD1@1% | IoU@25% | IoU@50% |
|---|---|---|---|---|---|---|
| A-SDF-GT [38] | 37.43 | 20.14 | 53.24 | 17.52 | 15.24 | 0.68 |
| A-SDF-GT-2 [38] | 38.23 | 21.05 | 53.58 | 19.34 | 18.66 | 0.57 |
| Ours-BG | 57.44 | 52.97 | 57.09 | **42.59** | 29.17 | 0.17 |
| Ours | **58.66** | **54.60** | **59.12** | 40.87 | **30.76** | **1.28** |

Table 13: Shape mAP result of storage category.

|  | Fscore@80% | Fscore@90% | CD1@5% | CD1@1% | IoU@25% | IoU@50% |
|---|---|---|---|---|---|---|
| A-SDF-GT [38] | 57.68 | 33.22 | 79.75 | 25.26 | **19.23** | **0.00** |
| A-SDF-GT-2 [38] | 57.57 | 32.08 | **81.11** | 25.48 | 16.38 | 0.00 |
| Ours-BG | 73.30 | **51.98** | 79.97 | **29.69** | 5.57 | 0.00 |
| Ours | **73.67** | 47.16 | 78.91 | 28.74 | 14.61 | 0.00 |

Table 14: Shape mAP result of table category.

|  | Fscore@80% | Fscore@90% | CD1@5% | CD1@1% | IoU@25% | IoU@50% |
|---|---|---|---|---|---|---|
| A-SDF-GT [38] | 84.62 | 63.59 | 89.74 | 59.23 | 72.69 | **12.95** |
| A-SDF-GT-2 [38] | **86.03** | **71.28** | **90.13** | **65.51** | **77.05** | 8.33 |
| Ours-BG | 64.42 | 63.02 | 65.78 | 63.00 | 59.50 | 4.94 |
| Ours | 65.81 | 64.05 | 69.12 | 64.07 | 57.62 | 5.40 |

Table 15: Shape mAP result of microwave category.

|  | Fscore@80% | Fscore@90% | CD1@5% | CD1@1% | IoU@25% | IoU@50% |
|---|---|---|---|---|---|---|
| A-SDF-GT [38] | 63.88 | 39.76 | 72.97 | 29.70 | 7.15 | 0.00 |
| A-SDF-GT-2 [38] | 71.03 | 52.85 | **75.88** | 44.48 | 13.82 | 0.12 |
| Ours-BG | **73.19** | **68.83** | 73.64 | **62.41** | **20.88** | 0.09 |
| Ours | 70.66 | 65.69 | 74.59 | 56.36 | 19.92 | **0.23** |

Table 16: Shape mAP result of refrigerator category.

|                  | Fscore@80% | Fscore@90% | CD1@5% | CD1@1% | IoU@25% | IoU@50% |
|------------------|------------|------------|--------|--------|---------|---------|
| A-SDF-GT [38]    | 65.85      | 41.10      | 77.38  | 24.41  | 77.72   | 48.04   |
| A-SDF-GT-2 [38]  | 67.64      | 41.10      | 75.48  | 22.84  | 79.06   | 48.38   |
| Ours-BG          | 86.40      | 78.12      | 87.43  | **45.74** | 87.22 | **73.74** |
| Ours             | **90.30**  | **83.25**  | **92.47** | 42.96 | **92.29** | 73.16 |

Table 17: Shape mAP result of washing machine category.

|                  | Fscore@80% | Fscore@90% | CD1@5% | CD1@1% | IoU@25% | IoU@50% |
|------------------|------------|------------|--------|--------|---------|---------|
| A-SDF-GT [38]    | 53.96      | 43.57      | 69.89  | 43.80  | 10.86   | 0.00    |
| A-SDF-GT-2 [38]  | 74.38      | 62.93      | **80.05** | 55.96 | **11.10** | **0.24** |
| Ours-BG          | **77.10**  | **74.12**  | 78.47  | **67.02** | 10.00 | 0.00    |
| Ours             | 74.74      | 69.19      | 77.57  | 62.03  | 8.96    | 0.00    |

Table 18: Shape mAP result of box category.

# I  Effect of the refiner on kinematic estimation

The effect of the refiner $\mathcal{R}$ on kinematic estimation is shown in Fig. 14. We conduct evaluations on joint parameter estimation. These evaluations focus on the intersection of true positive detections from both the model with and without the refiner with various mAP thresholds. Except for the joint axis orientation error at mAP@90%, the refiner improves the joint parameter estimation.

# J  Additional qualitative results on synthetic dataset

We show additional qualitative results on SAPIEN [58] dataset in Fig. 15.

# K  Additional qualitative results on real-world data

Additional qualitative results on real-world data are shown in Fig. 16. For the washing machine example in the top left corner of the figure, a semi-automatic segmentation tool on Microsoft PowerPoint was used for foreground segmentation by specifying the rough foreground region, as https://remove.bg could not properly segment the foreground in this example.

# L  Qualitatative results from novel viewpoint

Corresponding to Fig. 5 and Fig. 7 in 4, we visualize the qualitatative results from novel viewpoint on SAPIEN [58] dataset and BMVC [36] dataset in Fig. 17 and Fig. 18, respectively.

# M  Failure cases and limitations

**Uncertainty in closed parts**  One failure case arises when our model attempts to estimate the orientation of a part opening while it is in its closed state, as illustrated in Fig. 19 (a). We show quantitative evaluation on axis direction error for closed parts in Table 20. We regard a part as closed when its ground truth motion amount is less than a threshold. For revolute joints, we use $5°$ and $10°$ as thresholds, and 5% and 10% of the fully opened amount for prismatic joint. As seen in the table, the model successfully estimates reasonable joint direction for prismatic parts, while for revolute

|                  | Fscore@80% | Fscore@90% | CD1@5% | CD1@1% | IoU@25% | IoU@50% | Params. |
|------------------|------------|------------|--------|--------|---------|---------|---------|
| $N_\mathcal{D} = 16$ | 73.25  | 68.27      | 74.76  | **58.64** | **40.74** | **10.57** | 14.33M |
| $N_\mathcal{D} = 8$  | 71.88  | 65.38      | 73.68  | 54.79  | 35.31   | 8.20    | 9.05M   |
| $N_\mathcal{D} = 6, N_\mathcal{R} = 2$ | **74.22** | **68.80** | **75.71** | 58.61 | 40.06 | 9.80 | 10.25M |

Table 19: Ablation on the refiner $\mathcal{R}$ for shape mAP and the number of parameters.

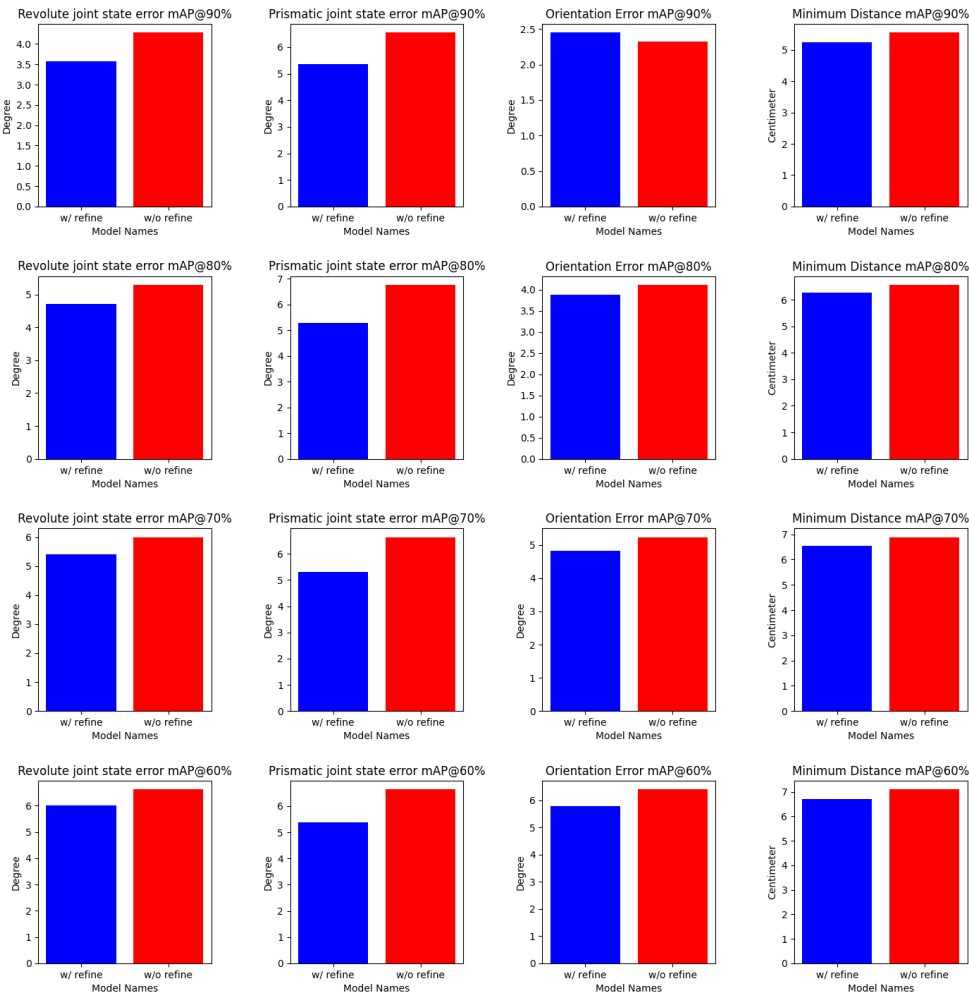

Figure 14: Effect of the refiner to joint parameter estimation.

| | <5°/5% | | | <10°/10% | | |
|---|---|---|---|---|---|---|
| | Revolute | Prismatic | Average | Revolute | Prismatic | Average |
| Ours | 25.96°(3.15°) | 2.04°(2.60°) | 24.46°(3.08°) | 18.09°(2.80°) | 2.37°(2.60°) | 16.79°(2.78°) |
| Ours-BG | 30.56°(3.39°) | 3.36°(2.42°) | 28.85°(3.27°) | 22.27°(2.90°) | 2.56°(2.43°) | 20.64°(2.85°) |

Table 20: Evaluation of joint axis direction error for closed parts. The part is regarded as closed if its ground truth motion amount is less than the threshold. For the prismatic part, we normalize the motion amount by its fully opened amount for thresholding to consider the different scales of parts. For reference, we denote the corresponding error of open parts in the parenthesis.

parts, the error is larger. Two contributing factors are: (1) our model does not explicitly model the uncertainty, particularly in ambiguous cases like the joint axis direction shown in Fig. 19 (a). The prismatic part's joint direction has less diversity due to the physical constraint by the base part, while several joint directions are possible for the revolute joint given the base part pose, thus the opening direction is ambiguous to the model. (2) The model could not detect small geometric cues from noisy depth maps, such as handles or knobs, which can provide valuable information about opening directions. These cases can be improved in several ways. For instance, we could extend the model to treat joint parameter ambiguity as a probability distribution, providing multiple joint parameter candidates sampled from the estimated distribution. This approach can still be practical, limiting the

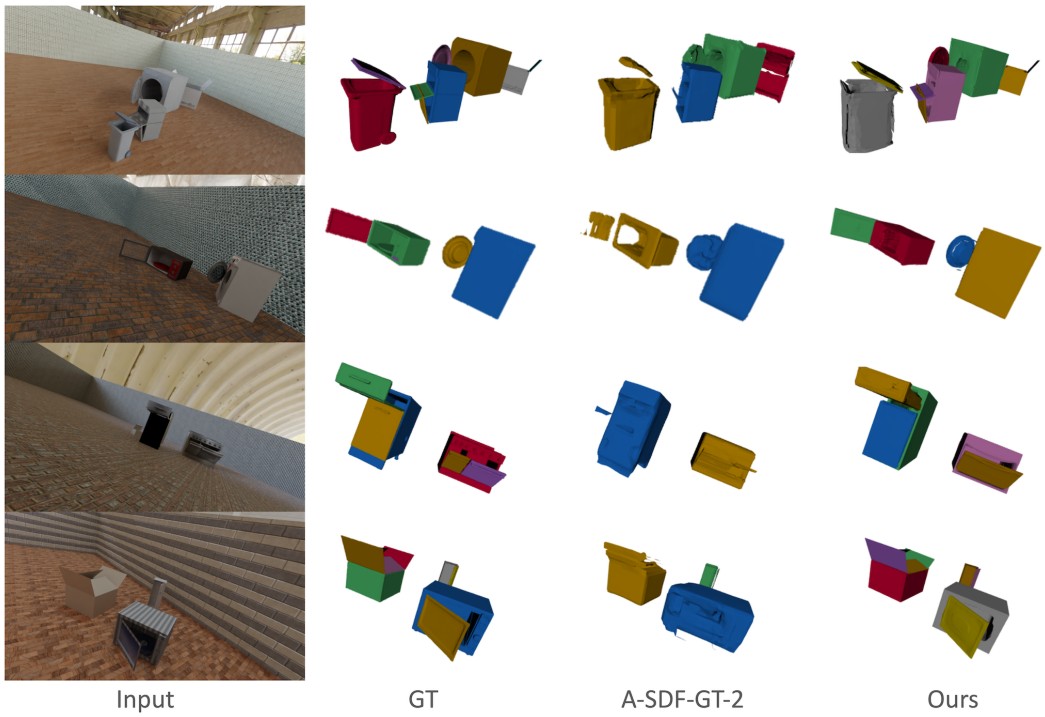

| Input | GT | A-SDF-GT-2 | Ours |

Figure 15: Additional qualitative results on SAPIEN [58] dataset.

kinematics parameter space to a few options for initial interaction steps in robotics applications or allowing humans to choose the best prediction in AR/VR applications during a human-in-the-loop process. Previous work [1, 11] has shown the effectiveness of this probability-based approach. Another approach would be to integrate 2D and 3D features into the detection backbone [54, 26, 7]. Incorporating 2D features would enhance the encoder's ability to capture small yet informative features, like handles or knobs, that may be missed by a sparse point cloud representation alone.

**Unseen categories**    Another failure case is that our model struggles to reconstruct instances from unseen categories, particularly those with shapes and part structures with small samples or absent in training data. Fig. 19 (b) shows the failure case with a dehumidifier as an unseen category, which has an unseen drawer shape and part structure; the base part only has one bottom drawer. This limitation can be improved by incorporating more data from recent datasets, such as [13, 34], which offers scanned real-world objects of various shapes, sizes, and part structures. However, it's important to note that generalization to unseen categories is beyond the scope of this paper. Moreover, in terms of part structure, it's also important to note that previous works can only handle very limited part structures, such as one base part and one articulated part [16] and predefined part count per category [38]. In contrast, our model can handle various part counts with a single model.

**False positives and negatives**    Due to the detection-based nature of our model and also the domain gap to real-world data, false negatives or positives still occur, as shown in Fig. 19 (c) and (d). One practical improvement would be including real-world data [13, 34] in training to minimize the domain gap, hence boosting the model's generalization capabilities. Another approach for reducing false positives especially with physically implausible configurations, is to include regularization losses, such as physical visolation loss [61] for learning physical constraints. It's important to note that the kinematics-aware part fusion improves both false positives and negatives, as discussed in Sec. 4.3.

**Objects with many joints**    The proposed approach works reasonably for complex articulated objects such as those with more than five joints, as shown in Fig. 20 (a) when all parts are visible from the given viewpoint. However, from certain viewpoints, we also observe that a single instance is reconstructed as two separate instances, each with fewer joints, as shown in the first row of Fig. 20

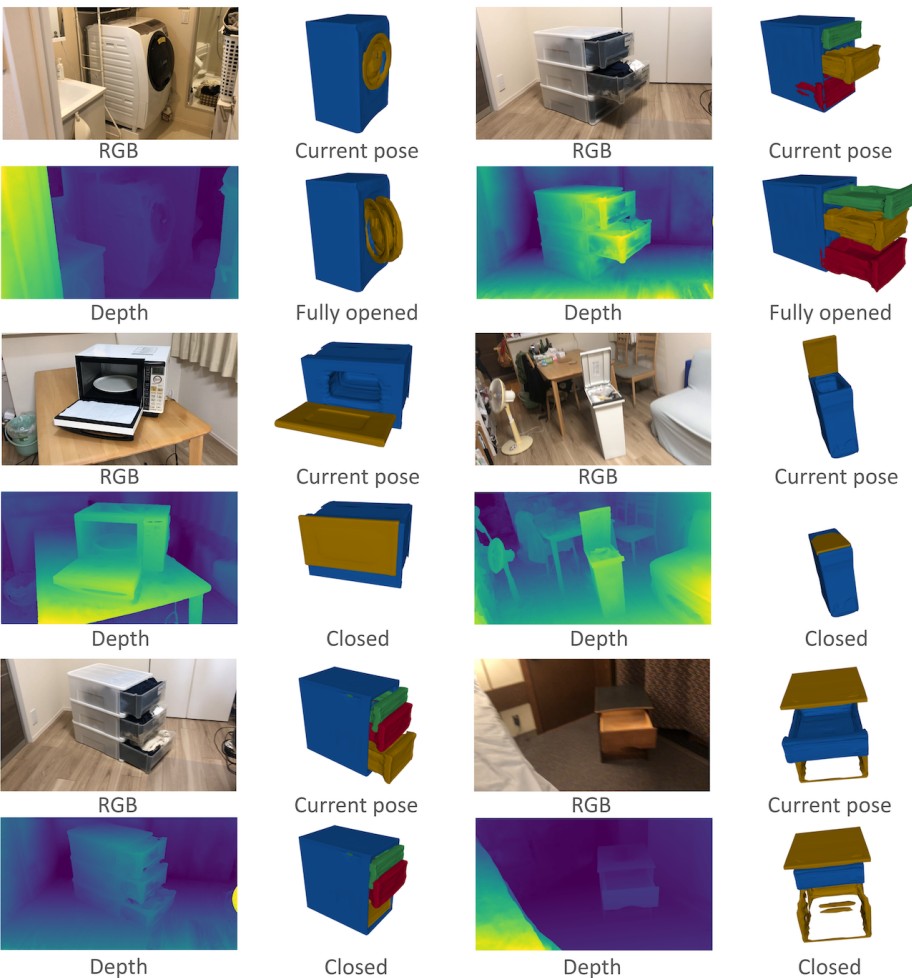

Figure 16: Additional qualitative results on real-world data.

(b). We attribute this to the current dataset consisting of a small number of CAD models with >5 joints, while and the majority of instances have fewer joints. Also, when some parts are only partially visible, our method tends to make inaccurate pose estimations for such parts, such as the right stacked three drawers indicated by the red box in the second row of Fig. 20 (b).

**Latency of KPF module**   While the KPF module improves detection performance, query oversampling, part proposal fusing, and kIoU calculation add extra running time for inference. We visualize the latency and accuracy trade-off in Fig. 21. We change the number of query oversampling from one to ten, and measure the per image processing speed in seconds. We randomly sample 100 samples from the test split for the evaluation and latency is measured on a V100 GPU. Although detection perfomance in shape mAP improves with more query oversampling, the latency degrades. Note that the result of w/o KPF applies 3D cuboid NMS with GPU parallelization, while the KPF module is implemented on CPU without any parallelization. We believe the latency of the KPF module can be significantly improved by optimizing the implementation, such as utilizing GPU parallization.

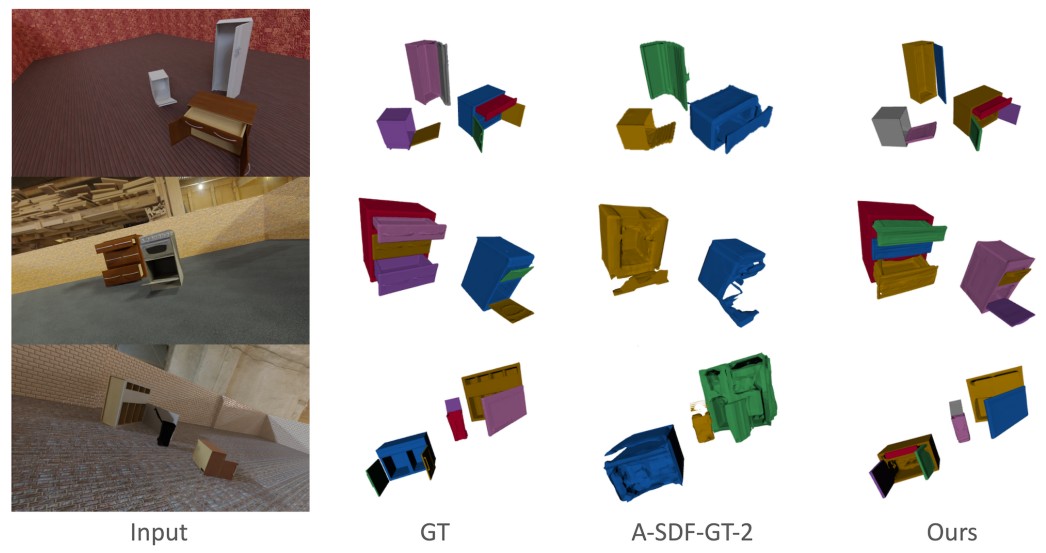

Figure 17: Qualitatative results on SAPIEN [58] from novel viewpoint.

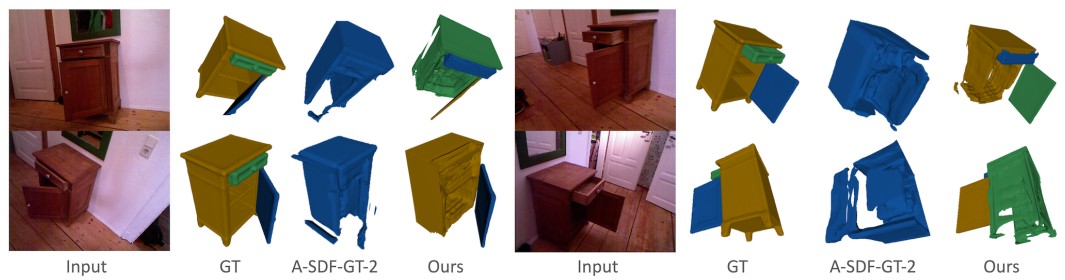

Figure 18: Qualitatative results on BMVC [36] from novel viewpoint.

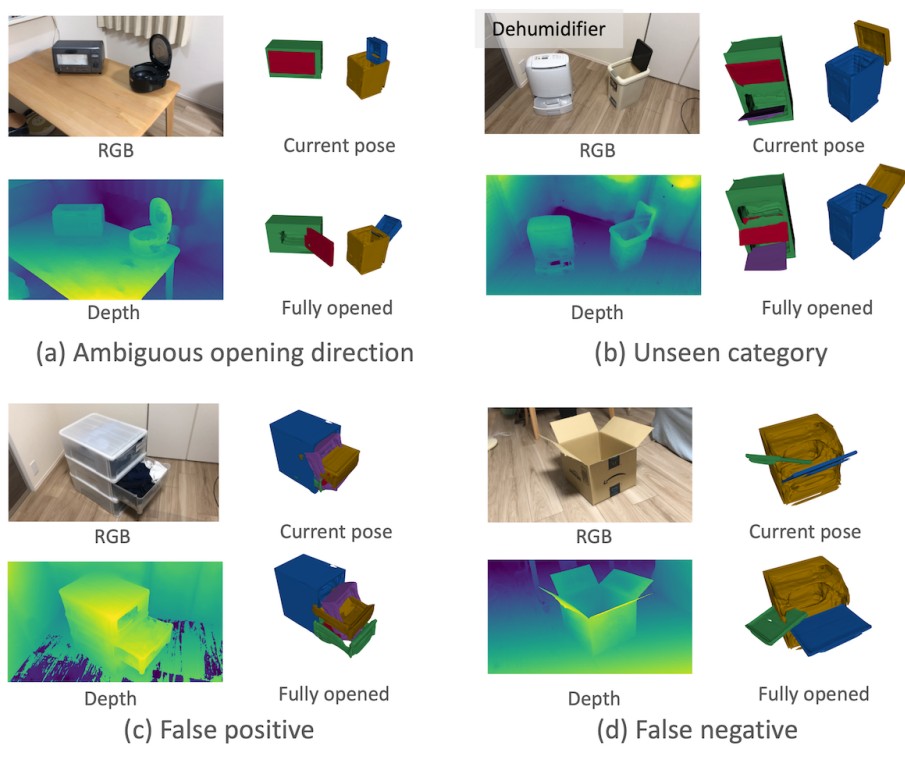

Figure 19: Failure cases on real-world data.

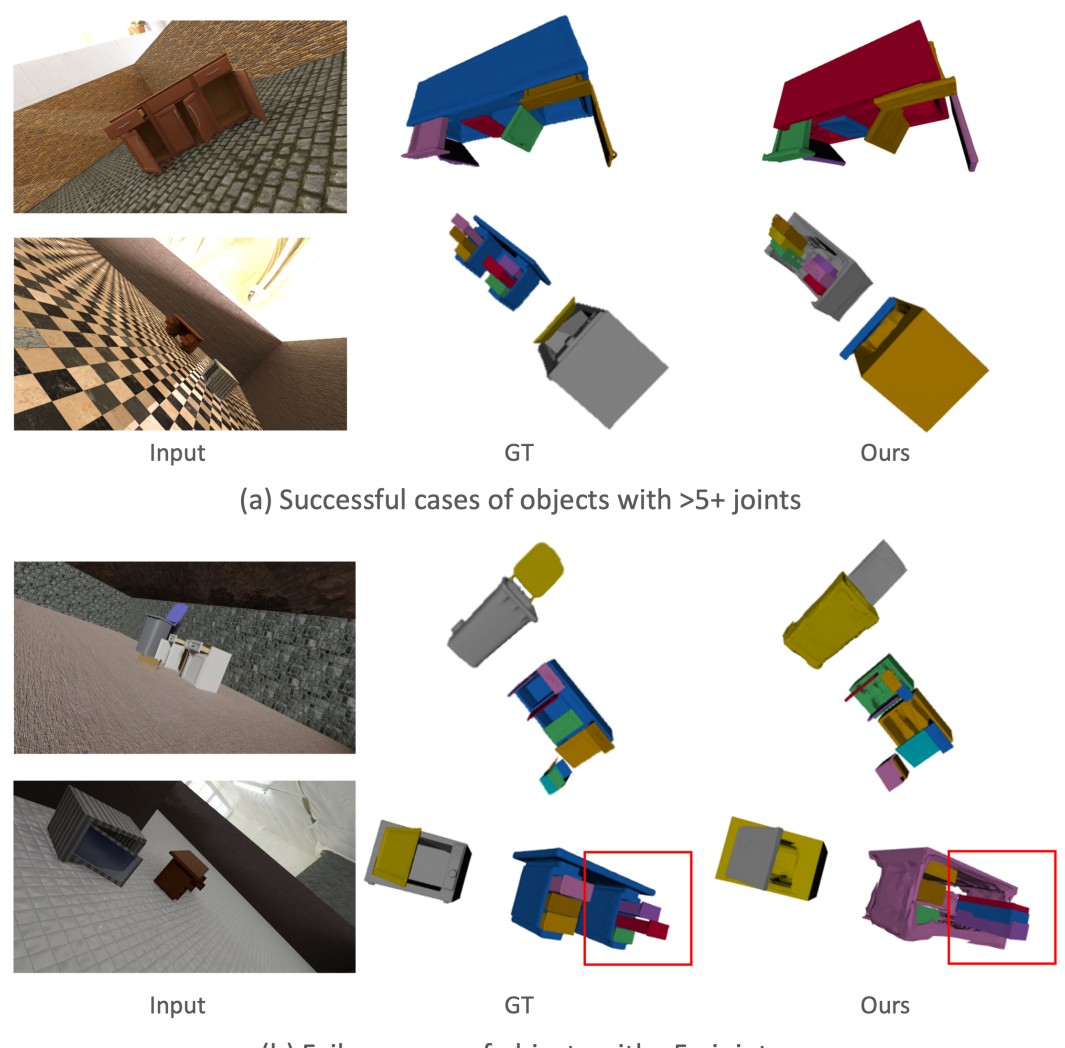

(a) Successful cases of objects with >5+ joints

(b) Failure cases of objects with >5+ joints

Figure 20: Successful and failure cases of articulated objects with more than five joints.

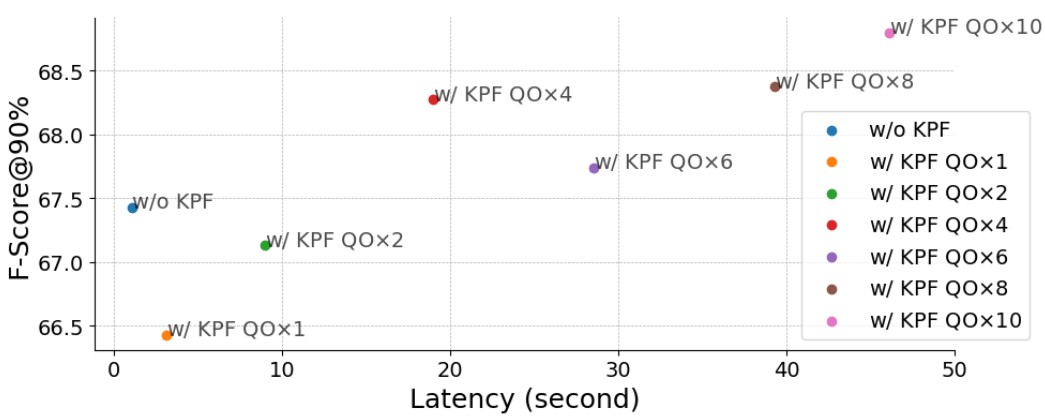

Figure 21: Latency and accuracy trade-off for different numbers of query oversampling (QO) in KPF module.

