# OpenReview forum: "Detection Based Part-level Articulated Object Reconstruction from Single RGBD Image"
_NeurIPS.cc/2023/Conference — NeurIPS 2023 poster_

### Official Review · Reviewer_mcMG · 2023-07-02

**Soundness:** 3 good
**Presentation:** 3 good
**Contribution:** 3 good
**Rating:** 6
**Confidence:** 4

**Summary:**

The paper presents a novel task focused on the reconstruction of multiple articulated objects, considering part-level shape, pose, joint parameters, and part-instance association, using only a single RGBD image. The authors propose an effective detect-and-group strategy that harnesses the part-level representations to detect, reconstruct, and predict parameters for articulated objects with diverse structures. Additionally, to enhance the detection performance, the paper introduces an oversampling and fusion strategy during inference. Moreover, the authors incorporate anisotropic size normalization and a refinement module to improve reconstruction quality and enhance part pose/motion prediction.

**Strengths:**

* The proposed task of reconstructing any number of articulated objects is both novel and valuable for exploring potential downstream robotics applications.
* Utilizing part-level representations and employing the detect-then-group strategy are intuitive and effective when handling articulated objects with diverse structures.
* The proposed end-to-end method builds upon 3DETR by incorporating part-level detection and pose/motion prediction. Furthermore, the authors introduce an instance-loss to aid in grouping within the part latent space.
* The experimental settings are reasonable and allow for meaningful comparisons with previous work in the fields of articulated object reconstruction and motion prediction. The results demonstrate the effectiveness of the proposed method.


**Weaknesses:**

* Due to the task's focus on reconstructing part shapes, it’s crucial to show some novel viewpoint of the recons trued shapes. For example, the reconstructed shapes of some base parts are hard to see the improvement only with quantitative results. Additionally, all visualizations in the paper appear to align with the input RGBD viewpoint, limiting the comprehensive understanding of the reconstructed shapes.
* If I understand correctly, for the statistics in table 2, the instance number includes the same articulated objects with various part states, or it’s hard to explain why there are so many instances in each category (much more than the object number in SAPIEN dataset)
* For the evaluation, it seems that in the test set, the part state are randomly initialized. Then most doors or drawers are actually open. However, in the real case, most time, the movable parts are closed. It’s better to have separate evaluation for the models with different motion states
* The paper lacks statistics on the number of articulated objects present in the input image, and it would be intriguing to include evaluation results that consider the number of objects to better understand the performance of part-instance association.
* The quantitative improvements achieved with QO, PF, and kIoU metrics do not appear to be substantial. Are there any qualitative results available to provide additional insights or demonstrate the effectiveness of the proposed module?


**Questions:**

* Why the number of images in the validation set is much more than the number of images in the train set and test set?
* For the number of objects in each image, what’s the statistics in the train set and test set?
* For the model, why “ours-BG” sometimes outperform “ours”? “Ours” should have extra information of the foreground mask, right? Is this caused by the imperfect performance of the foreground mask?


**Limitations:**

The authors mention the limitation in the supplement.

---

> ### Author Rebuttal · Authors · 2023-08-10
>
> Thank you for your feedback and questions!
>
> ## Novel viewpoint of the reconstructed shapes
> We promised to add novel views in camera ready. For reference, we show the novel viewpoint of the reconstructed shape in Fig. 5 and Fig. 7 of the main paper in Fig. 4 of the attached material. Our method qualitatively better reconstructs the shape of occluded regions than the baseline.
>
>
> ## The instance number in Table 2
> We augment each CAD model from the SAPIEN dataset in terms of three side lengths and randomize the part poses as explained in Section 4, L230. Randomizing the three side lengths changes the original shape significantly. Thus we count it as a unique instance in Table 2. We list the number of original CAD models for each category per split in Table 1 of the appendix. We also show the size distribution after the augmentation in Fig. 1 of the appendix.
>
>
> ## Evaluation for closed parts
> We follow the relevant previous works’ [1,2,3,4,5] standard experimental settings of uniformly randomized articulation for the test set. We qualitatively confirmed that our approach reasonably works for closed or nearly closed parts, as visualized in Fig. 7 (right) of the main paper, Fig 6. (top row) of the appendix, and Fig 3 (bottom, three stacked drawers on the left) of the attached material. We will add further analysis regarding the closed parts in the camera ready.
>
>
> ## Evaluation results that consider the number of instances
> We have evaluated shape mAP in terms of different numbers of instances in a scene, as shown in Fig. 5 of the attached material. As the number of instances in a scene grows, F-Score decreases. We attribute this to the camera being often distant from instances when a view contains multiple instances. Thus it becomes harder to understand the fine details of the part geometry accurately.
>
>
>
> ## Qualitative results for the KPF module
> Please refer to the global comment for the detail.
>
> ## Larger validation set than the training set
> Also, please refer to the global comment for the detail as well.
>
>
> ## Statistics of the number of objects in each image
> The average number of instances for the training set is 2.154, and 2.158 for the test set.
>
>
>
>
>
>
> ## Why “Ours-BG” sometimes outperform “Ours”?
> As pointed out, the imperfect performance of the foreground mask can be one reason. Another reason could be that sometimes background context near the instance would help detect an instance and estimate the part pose. In some cases, the background context of the floor might be informative for estimating the rotation of the part pose based on the tilt of the floor, and the position of the floor might help to estimate the center of the part. That being said, "Ours" without background outperforms "Ours-BG" with background in the majority of the metrics, as in the majority of cases when there is a background, less number of queries cover the foreground objects, becoming disadvantageous for detection and detailed understanding of part shapes and poses.
>
> [1] Heppert et al. CARTO: Category and Joint Agnostic Reconstruction of ARTiculated Objects, CVPR 2023.
>
> [2] Liu et al. AKB-48: A Real-World Articulated Object Knowledge Base, CVPR 2022.
>
> [3] Kawana et al. Unsupervised Pose-aware Part Decomposition for Man-made Articulated Objects, ECCV 2022.
>
> [4] Jiang et al. OPD: Single-view 3D Openable Part Detection, ECCV 2022.
>
> [5] Mu et al. A-SDF: Learning Disentangled Signed Distance Functions for Articulated Shape Representation, ICCV 2021.

---

> > ### Comment · Reviewer_mcMG · 2023-08-17
> >
> > Thanks to the response from the authors. The rebuttal has resolved my questions.

---

### Official Review · Reviewer_HLDh · 2023-07-04

**Soundness:** 3 good
**Presentation:** 3 good
**Contribution:** 2 fair
**Rating:** 5
**Confidence:** 4

**Summary:**

This paper presents a new method for 3D semantic instance reconstruction at the part level from a single RGB-D image. This method follows a top-down manner by first detecting object parts using 3DETR, where each instance part's bounding box will be predicted. The point cloud located inside each bounding box contains the part surface geometry, which will be normalized into a uniformly-scaled canonical system for the following part shape reconstruction. The key contribution of this paper is the kinematics-aware part fusion module, which follows a bottom-up manner to constitute complete instance shapes from parts.

The paper writing could be improved, e.g., the first paragraph in the introduction section is actually elaborating on related work. An illustration in Sec 3.6 would be more accessible for the audience to follow.

The experiment metrics and designs are extensive but lack some comparisons with the state-of-the-art.


**Strengths:**

In my view, the major strengths can be concluded as follows:

1. Kinematics-aware part fusion (KPF). The authors follow a top-down and bottom-up manner to first detect part-level geometries from the point cloud. Then they use KPF to construct instance shapes from parts. These two processes are trained end-to-end jointly to improve the entire performance.

2. It shows good qualitative and quantitative performance in both synthetic data and synthetic data, with an acceptable generalization ability, even though they only train their model with synthetic data.

**Weaknesses:**

The weaknesses of this paper are also obvious.

1. This paper follows a similar top-down pipeline as previous works. (e.g., RfD-Net). In L34, the authors argued that previous methods "are either limited to a single instance or require a separate instance detector, making them not end-to-end trainable". I do not think this makes sense, as this paper also requires an instance detector. Besides, relying on a separate instance detector does not mean they can not be trained end-to-end.

2. As this task is not new and this paper follows a similar pipeline, it means the contribution of this paper only comes from the module design. To me the major contribution lies on KPF module. But from the ablation study, it seems KPF does not contribute the most.

3. Section 4.2, comparing with OPD by adjusting the IoU threshold is not rigorous. There is no theoretical guarantee that this comparison is fair. Since OPD is an RGB-based method while this paper uses RGB-D information, a fair comparison would be to add depth information into OPD or to remove the depth channel in your method.

**Questions:**

Here are some suggestions to improve this work, with which I would consider raising my score.

1. A comprehensive analysis and comparison between the key module of this method (e.g., KPF) and the state-of-the-art (A-SDF is a bit old).
2. The introduction should be about telling the story of the paper's motivation, key observations and contributions. Listing many related works there made it difficult to follow.

**Limitations:**

This paper well discussed the limitations and failure cases in the supplemental.

---

> ### Author Rebuttal · Authors · 2023-08-10
>
> Thanks for your detailed review!
>
> ## Effectiveness of the KPF module and comprehensive analysis
> The primary source of improvement in shape reconstruction accuracy is our part-level reconstruction approach enabling the reconstruction of articulated objects with various part counts, which is the main contribution of this paper. In addition, we also observed qualitative improvement using the KPF module for challenging targets like distant and small part shapes, which we detailed in the global comment.
>
> ## Comparison with more recent state-of-the-art baseline
> We have added another baseline based on AKBNet from CVPR 2022 as a more recent state-of-the-art method besides A-SDF, as detailed in the global comment.
>
> ## Improving introduction
> We show the revised introduction idea below and promise to revise the introduction further in the camera ready.
> ```
> Estimating object shape, pose, size, and kinematics from a single frame of partial observation is a fundamental challenge in computer vision. Understanding such properties of daily articulated objects like refrigerators and drawers has various applications in robotics and AR/VR.
>
> Shape reconstruction of daily articulated objects is a challenging task. First, the objects have various shapes resulting from different local part poses. More importantly, they have intra- and inter-category diverse part configurations regarding part counts and structures. A combination of those factors results in exponentially increasing shape variation. Previous works handles only fixed part count by a single model [1] or use multiple category-level models for different part counts after instance-level detection [2] by modeling the target shape in instance-level latent space. Handling those varieties with a single model is a complex and unsolved task.
>
> In this paper, we address this complexity through the novel detect-then-group approach. Our key observation is that the daily articulated objects consist of similar part shapes. For example, regardless of the number of refrigerator doors, each door can have a similar shape, and the base part may have similar shapes to those from other categories, such as storage. Detecting each part and then grouping them into multiple instances is a scalable and generalizable approach for diverse part configurations of daily articulated objects in a scene.
>
> Based on this idea, we propose an end-to-end detection-based approach for part-level shape reconstruction. <the rest of the text will follow the second and the third paragraphs of the introduction>
> ```
>
> ## Improving the figure for the KPF module
> We promise to revise the current figure in Section 3.6 in the camera ready for better accessibility.
>
> ## Using an instance detector does not mean end-to-end trainable
> We agree that some methods, like RfD-Net, operate end-to-end from detection to reconstruction. Therefore, we promise to rephrase the original sentence not to imply that detection-based necessarily means not end-to-end trainable up to shape reconstruction.
>
> ## Contribution of the paper
> As a high-level idea, we base our approach on detector-based reconstruction approaches. However, we have developed these ideas and made novel progress. As reviewers **mcMG** and **MJHC** agree, introducing a detect-then-group approach that can handle daily articulated objects with an arbitrary number of parts, a major limitation of previous works, is novel. This presents a new setting for shape reconstruction of daily articulated objects and stands as the significant contribution of this paper against the recent previous works.
>
> ## Fairness of comparison against OPD
> As written in L275 of the main paper, we have included depth alongside RGB as an additional channel for OPD's input, ensuring a fair comparison. Addressing concerns about the IoU threshold's fairness, we've also conducted an additional evaluation with a favorable setting for OPD, which is presented in the table below. While we maintained the same evaluation settings as in the main paper, we selected the pair of IoU threshold values for OPD and ours from all possible combinations (50% to 90% with 10% steps for both OPD and ours) this time. We chose a pair of IoU thresholds in a way that gave the best metric values for OPD. In the table below, IoU thresholds for OPD and ours are 70% and 80% for the prismatic joint state, respectively, and 90% for the rest of the joint parameters for both OPD and ours. Despite these adjustments, our method still outperforms OPD.
>
> ### Additional joint state evaluation
> |       | State (revolute/prismatic)                     | OE         | MD     |
> |-------|---------------------------|---------------|--------------|
> | OPD   | 16.52°/16.46cm            | 10.81°        | 34.68cm      |
> | Ours-BG | **3.34°**/**5.45cm**    | **1.96°**     | **5.15cm**   |
>
>
>
> [1] Heppert et al. CARTO: Category and Joint Agnostic Reconstruction of ARTiculated Objects, CVPR 2023.
>
> [2] Liu et al. AKB-48: A Real-World Articulated Object Knowledge Base, CVPR 2022.

---

> > ### Comment · Reviewer_HLDh · 2023-08-17
> > **Post rebuttal comments**
> >
> > Thanks to the authors for their comprehensive rebuttal. I believe the authors addressed my concerns, and would like to raise my score to weak accept.

---

### Official Review · Reviewer_MJHC · 2023-07-05

**Soundness:** 3 good
**Presentation:** 3 good
**Contribution:** 3 good
**Rating:** 6
**Confidence:** 4

**Summary:**

The paper presents a detection-based reconstruction method for articulated objects along with estimating part-level 6D object poses, sizes, and joint parameters. The paper uses 3DETR as the backbone predicts all of the above quantities while treating this problem as a supervised learning approach given labeled synthetic data. The approach trains only on synthetic data and transfers to real world dataset.

**Strengths:**

The paper effectively uses a detection backbone i.e. 3DETR to reconstruct multiple articulated objects from a single view RGB-D observation. Sim-to-real transfer uses only synthetic data pretraining is a strong result. The use of set matching (from instance segmentation) literature for articulated part reconstruction is interesting. Joint estimation and shape reconstruction quantitative results are significantly better than baselines. I believe breaking the problem in this way i.e. part level reconstruction omits the need for model selection (i.e. training one model per category such as different models for glasses (2 joints) and refrigerators (1 joint) which is a major limitation in previous works.

**Weaknesses:**

1. The paper mentions CARTO, which is super relevant and recent CVPR'23 work but doesn't directly compare to it in terms of both detection and reconstruction. Is there a reason for it? Rather the paper takes individual joint parameter baseline i.e. OPD and shape reconstruction baseline i.e. A-SDF and compares them separately to those. While these are relevant, CARTO is the most relevant to this work in terms of system-level joint detection and shape reconstruction.

2. While the work offers a solution for handling an arbitrary number of parts in the image, the qualitative results show simple examples i.e. same types of joints and 2 joints mostly. Did the authors test their approach on more complicated geometries i.e. varying number of joints or articulated objects with more joints >5+.

3. I didn't see a discussion to run the speed of the model. The most relevant works which the author discusses i.e. CARTO (which builds upon CenterSnap and ShaPO) are all fast approaches. Is 3DETR equally fast and did the author consider adding that backbone i.e. CenterSnap backbone which offers a faster solution + equally good model in terms of accuracy? This is crucial for real-time applications like robotics or grasping etc.

**Questions:**

Please see the questions raised above in the weakness section. To summarize, a comparison to strong and relevant system-level baselines, more qualitative results, and discussion/comparison to speed vs accuracy tradeoff would be useful to answer.

**Limitations:**

The authors can only handle cases where instance boundaries are defined so it would fail in scenarios where a door would be attached to a room let's say.

---

> ### Author Rebuttal · Authors · 2023-08-10
>
> Thank you for your feedback!
>
> ## Targeting articulated objects with >5+ joints
> The trained model works reasonably for complex target instances like >5+ joints, as shown in Fig. 2 of the attached material, especially when all parts are clearly visible from the given view. However, from a certain viewpoint, we also observe that the single instance is reconstructed as two separate instances (Fig.3 of the attached material, first row) with fewer joints for each instance. We attribute this to the current dataset consisting of a small number of CAD models with >5+ joints, and the majority of instances have fewer joints. Also, when some parts are only partially visible, our method tends to make inaccurate pose estimations for such parts (Fig.3 of the attached material, second row, the right stacked three drawers indicated by red box). We add these cases as limitations in camera ready.
>
> ## Comparison with CARTO
> Please refer to the global comment on the comparison with state-of-the-art.
>
> ## Real-time application
> Currently, improvement in detection-to-reconstruction speed compared to the state-of-the-art methods is not our focus in this paper. Combination of real-time DETR-based 3D detectors like MonoDTR [1], tuning the number of queries based on speed vs. accuracy trade-off, employing hierarchical isosurface sampling for fast meshing as CARTO, re-implementing current single processed, CPU-based implementation of KPF module with multiprocessed or GPU-based implementation can be possible extensions for speed improvement, and we leave it for future work. We will include quantitative speed measurement in the camera ready.
>
> [1] Huang et al. MonoDTR: Monocular 3D Object Detection with Depth-Aware Transformer, CVPR 2022.

---

> > ### Comment · Reviewer_MJHC · 2023-08-15
> > **Post rebuttal comments**
> >
> > The rebuttal has resolved my questions. Adding limitation examples, and quantitative comparisons to existing state-of-the-art fast reconstruction techniques like CARTO/ShAPO (and mentioning this as a potential limitation as well) would strengthen the arguments in the paper and I look forward to seeing that in the final version. I am happy with the novelty of the paper i.e. part based reconstruction to handle multiple joint types and retain my rating.

---

### Official Review · Reviewer_Y464 · 2023-07-07

**Soundness:** 3 good
**Presentation:** 3 good
**Contribution:** 3 good
**Rating:** 6
**Confidence:** 4

**Summary:**

The paper proposes a method for man-made articulated objects reconstruction from a single RGBD image across different object categories. The method is based on a detect-then-group pipeline, using kinematics-aware fusion for addressing false negatives.


**Strengths:**

The proposed method addresses the challenging problem of man-made articulated object reconstruction from a single RGBD image, estimating at the same time each part's pose.

The overall architecture has a somewhat complicated structure with numerous encoder and decoder modules employed. Nevertheless, the design choices are sufficiently motivated in the text and the loss function is quite straightforward given the overall architecture. The experimental evaluation considers both a synthetic (SAPIEN) as well as a real-world (BMVC) dataset. The proposed method achieves improved performance with respect the state-of-the-art on SAPIEN. The ablative study shows the relative contribution of important components.

Regarding reproducibility, sufficient details are provided making it easier to reproduce the results (considering also the supplemental material).

**Weaknesses:**

The method seems to be closely related to [20]. I think a more detailed discussion about the relation between the two methods should be included in the Related work section. Related to this, it is not clear why the experimental evaluation does not consider comparison also with [20]. This is important, considering the few baselines available for this task. Alternatively, methods considering multi-view reconstruction on SAPIEN could also be considered to give additional insight on the performance of the proposed method. On a side note, the considered metrics could also include the joint state error.

### Minor comments
- L.45: define acronym NMS
- L.127: not clear
- Fig. 2 is not referenced in text
- Although prior work regarding human subjects is briefly visited in the related work, similar work for animals is briefly mentioned but not specified. Works like [R1], [R2] and [R3] could be included to make this aspect of related work stronger.

[R1] Ntouskos, V., Sanzari, M., et al. (2015). Component-wise modeling of articulated objects. ICCV

[R2] Zuffi, S., Kanazawa, A., et al. (2017). 3D menagerie: Modeling the 3D shape and pose of animals. CVPR

[R3] Jiang, L., Lee, C., Teotia, D., & Ostadabbas, S. (2022). Animal pose estimation: A closer look at the state-of-the-art, existing gaps and opportunities. Computer Vision and Image Understanding, 103483.

**Questions:**

- L.126: why P_O both conditions O and is given also as an argument?

**Limitations:**

Limitations are discussed in the text

---

> ### Author Rebuttal · Authors · 2023-08-10
>
> Thank you for your review and questions!
>
> ## Joint state error
> We show the joint state error compared with OPD in Table 3 in the main paper, denoted as “State.” We have also added the joint state error comparison against A-SDF-GT-2 in the table below. Note that A-SDF-GT-2 can only be evaluated against GT instances with its learned part counts. Therefore, we evaluated A-SDF-GT-2 and ours only on the GT instances with the part counts that A-SDF-GT-2 learned.
>
>
> ### Joint state error evaluation
> |               | Revolute (deg) | Prismatic (cm) |
> |---------------|----------------|----------------|
> | A-SDF-GT-2    | 25.49          | 13.97          |
> | Ours          | **4.62**       | **4.80**       |
>
>
>
>
> ## Why is P_O both conditions O and is also given as an argument?
> We appreciate the reviewer for pointing this out. Actually $P_\mathcal{O}$ is in the right place but $\mathbf{p}$ should be $\mathbf{x}$. The correct equation is as follows:
>
> $o_{\mathbf{x}} = {\mathcal{O}}((\mathbf{R}\mathbf{S})^{-1}(\mathbf{x}-\mathbf{c}) \mid P_{\mathcal{O}},\mathbf{h})$
>
> $\mathbf{x}$ is in world coordinates, and $(\mathbf{R}\mathbf{S})^{-1}(\mathbf{x}-\mathbf{c})$ projects it into the local coordinate of the part to sample occupancy value at world coordinates $\mathbf{x}$. We will fix the equation in camera ready.
>
> ## Comparison with PPD [1]
> Please refer to the global comment onthe comparison with state-of-the-art.
>
> ## Additional related works
> We appreciate an idea to improve our related work section. We will include them in the camera ready.
>
> ## Writing errors
> As responded in the global comment, we promise to fix this in camera ready.
>
> [1] Kawana et al. Unsupervised Pose-aware Part Decomposition for Man-made Articulated Objects, ECCV 2022.

---

> > ### Comment · Reviewer_Y464 · 2023-08-16
> >
> > I thank the reviewers for the detailed reply and the clarifications provided in their rebuttal. I have no further questions or comments at this time.

---

### Official Review · Reviewer_VtGS · 2023-07-09

**Soundness:** 4 excellent
**Presentation:** 4 excellent
**Contribution:** 3 good
**Rating:** 8
**Confidence:** 3

**Summary:**

The paper proposes an end-to-end trainable method for reconstructing multiple articulated objects from a single RGB-D image, consisting of detecting parts, reconstructing part-level shapes, and estimating poses, bounding boxes as well as kinematic parameters. The parts are grouped into instances later. The authors propose anisotropic scale normalization for shape reconstruction to accommodate various part sizes and scales. Besides, the authors also propose test-time kinematic-aware part fusion (similar to non-maximum suppression) to reduce false positives when multiple detected results are generated and needed to be merged. Evaluation on both synthetic and real data demonstrates the effectiveness of the proposed method.

**Strengths:**

- The paper is clearly written and easy to follow.
- The authors propose a clean pipeline for reconstructing articulated objects that can be trained end-to-end and inferred straightforwardly without composing multiple networks.

**Weaknesses:**

Minor typos:
- L38: "a end-to-end" -> "an end-to-end"
- L219-L220: "and use cosine scheduler to learning rate of 1e-6" sounds a little strange to me.

**Questions:**

1. How do the authors choose the revolute origin? It seems that any point on the revolute axis can be chosen as the origin. I think this question also applies to prismatic joints.
2. Is there any explanation why anisotropic scaling is better than isotropic one?
3. L235-L236: Do you actually mean 20,000 validation images and 168,726 training images? Otherwise, it is strange why more images are used for validation instead of training.

**Limitations:**

The authors has addressed the limitations.

---

> ### Author Rebuttal · Authors · 2023-08-10
>
> Thanks for reviewing our paper!
>
> ## How to choose revolute origin?
> As pointed out, any point on the line formed by GT revolute origin and GT joint axis is the correct revolute origin. Thus for evaluation, we measure the minimum distance (MD) between the GT axis line and the predicted revolute origin. During training, we simply minimized the L2 distance between the GT revolute origin and prediction, following OPD. As a prismatic joint does not have a GT origin, we only optimize the predicted revolute origin if the matched GT part’s kinematic class is revolute type.
>
> ## Why is anisotropic scaling better than isotropic one?
> Although we have explained the reason in the main paper's L308 in Section 4.3 and L121-122 in Section 3.3 of the main paper, let us recap here again for better understanding. With isotropic scaling, the shape decoder needs to learn variations in shapes with different width, height, and depth ratios. Anisotropic scaling normalizes all three sides to unit length, reducing the variation of shapes that the shape decoder needs to learn and making it easier for the shape decoder to be optimized during the training.
>
>
> ## Validation data size
> Please refer to the global comment for a detailed response.
>
> ## Writing errors
> We appreciate the reviewer for pointing this out. As responded in the global comment, we promise to fix this in camera ready.

---

> > ### Comment · Reviewer_VtGS · 2023-08-13
> >
> > The rebuttal has resolved my questions. I would like to keep my rating.

---

### Author Rebuttal · Authors · 2023-08-10

# Global comment
We thank all the reviewers for their thoughtful feedback. We are encouraged that the reviewers having identified our paper making a good contribution (**mcMG**, **Y464**, **MJHC**, **VtGS**), and the proposed to be interesting (**MJHC**), intuitive and effective (**mcMG**), working on a novel task (**mcMG**), and addressing the major limitations of previous works (**MJHC**). We are also glad that all reviewers agree that the experiments demonstrate effectiveness and good performance on both synthetic and real data, significantly better than the baseline (**MJHC**), with reasonable experimental settings for meaningful comparisons with previous works (**mcMG**). Below, we write responses to the concerns common to several reviewers.


## Comparison to the state of the art (**Y464**, **MJHC**, **HLDh**):
To the best of our knowledge, no prior work operates in exactly the same setting as ours, which can handle cross-category, multiple-articulated objects with various part counts and structures. The closest work is the system-level (handling multiple articulated objects) reconstruction approach CARTO [1], which targets cross-category multiple articulated objects with a **single** articulated part. We were unable to directly compare against it due to the lack of an official code at the time of submission, and it was only made publicly available several days ago.

To alleviate this problem, we used the SOTA system-level baseline setting from the paper [1]: A-SDF with the ground truth (denoted as A-SDF-GT in our paper) for detection. This baseline provides an upper-bound system-level performance in detection and is used as a comparable baseline against CARTO in the paper [1]. To address concerns from fellow reviewers to consider another baseline (**Y464**) and to compare against a more recent approach than A-SDF from ICCV 2021 (**HLDh**), we added the new baseline based on the idea from AKBNet [2] (CVPR 2022), denoted as AKBNet-GT in the table below. AKBNet uses A-SDF for shape reconstruction and improves the accuracy of shape reconstruction by using the motion amount estimated by an additional, improved pose encoder during shape reconstruction. As the official AKBNet code for the encoder has not been released, we use the ground truth motion amount during evaluation instead. As it uses a category-level shape decoder, we use the same setting with A-SDF-GT-2 in the main paper that trains up to two most frequent part counts per category. Our approach still outperforms AKBNet-GT in the majority of metrics.

To avoid unfair comparison, we did not include another recent work, PPD [3], as our baseline, suggested by fellow reviewer **Y464**. This is because PPD focuses explicitly on shape **abstraction** but not accurate shape reconstruction with unsupervised learning. In contrast, our approach is fully-supervised, and targets shape reconstruction.



### Shape mAP evaluation
|               | Fscore@80% | Fscore@90% | CD1@5%   | CD1@1%   | IoU@25% | IoU@50% |
|---------------|------------|------------|---------|---------|--------|--------|
| AKBNet-GT   | 72.67      | 58.73      | **79.17** | 49.92   | **41.61** | 11.26  |
| Ours          | **74.77**   | **68.38**      | 77.39   | **56.53**   | 41.35  | **11.63** |



## Effectiveness of the KPF module (**mcMG**, **HLDh**):
We have detailed the effectiveness of the KPF module qualitatively through additional ablation studies in Appendix H. The KPF module improves detection and pose estimation, especially for small and distant parts, as visualized in Fig. 3 in the appendix while suppressing false positives, as shown in the precision score in Table 4 of the main paper. As explained in L159-161 in Section 3.6 of the main paper, the reason for this improvement is that the query oversampling allows more queries to cover better small parts represented as a small number of points in the input point cloud.

We added qualitative visualization of the KPF module in Fig. 1 of the attached material. We observe that the KPF module also improves detection for occluded parts with a small number of points in the input point cloud due to the occlusion (indicated by the bottom red box in the figure). Furthermore, we have qualitatively demonstrated the effectiveness of kIoU in the KPF module in Fig. 4 of the appendix. Query oversampling (QO) and part fusion (PF) alone with standard 3D box IoU results in false positives for thin parts like doors due to the small overlap between parts, but kIoU effectively suppresses the false positives by considering their overlapping trajectories, as explained in L171 in Section 3.6.




## Larger number of the validation set than the training set (**mcMG**, **VtGS**):
We generated 188,726 images for training and validation purposes. Due to our limited computational resources and time budget for the experiments in parallel, we used only 20,000 images for training. One can include the remaining images in the validation set to the training set to make the sizes of the validation and training sets comparable, depending on available computational resources and time budget.


## Writing improvement: typos, grammatical errors, missing reference to the figure, unclear sentence (**VtGS**, **Y464**):
We appreciate the reviewers pointing out the areas to improve in the manuscript. We promise to fix those issues in the camera-ready version.

[1] Heppert et al. CARTO: Category and Joint Agnostic Reconstruction of ARTiculated Objects, CVPR 2023.

[2] Liu et al. AKB-48: A Real-World Articulated Object Knowledge Base, CVPR 2022.

[3] Kawana et al. Unsupervised Pose-aware Part Decomposition for Man-made Articulated Objects, ECCV 2022.

---

### Decision · Program_Chairs · 2023-09-21

**Decision:**

Accept (poster)

**Comment:**

In this paper, the authors propose a new method for part-level articulated object reconstruction from single RGBD images. After rebuttal, all reviewers find this paper making solid contribution to our field and recommend acceptance. The AC agrees with the assessment.